# TNFAIP8 controls murine intestinal stem cell homeostasis and regeneration by regulating microbiome-induced Akt signaling

Jason R. Goldsmith [1✉], Nina Spitofsky[1], Ali Zamani[1], Ryan Hood[1], Amanda Boggs[1], Xinyuan Li[1], Mingyue Li[1], Elizabeth Reiner[1,2], Arshad Ayyaz[3], Zienab Etwebi[1], Ling Lu[1], Javier Rivera Guzman[1,4], Mayassa J. Bou-Dargham [1], Terry Cathoupolis[1], Hakon Hakonarson [5,6], Honghong Sun[1], Jeffrey L. Wrana[3,7], Michael V. Gonzalez[5,6] & Youhai H. Chen[1✉]

The intestine is a highly dynamic environment that requires tight control of the various inputs to maintain homeostasis and allow for proper responses to injury. It was recently found that the stem cell niche and epithelium is regenerated after injury by de-differentiated adult cells, through a process that gives rise to Sca1+ fetal-like cells and is driven by a transient population of Clu+ revival stem cells (revSCs). However, the molecular mechanisms that regulate this dynamic process have not been fully defined. Here we show that TNFAIP8 (also known as TIPE0) is a regulator of intestinal homeostasis that is vital for proper regeneration. TIPE0 functions through inhibiting basal Akt activation by the commensal microbiota via modulating membrane phospholipid abundance. Loss of TIPE0 in mice results in injury-resistant enterocytes, that are hyperproliferative, yet have regenerative deficits and are shifted towards a de-differentiated state. $Tipe0^{-/-}$ enterocytes show basal induction of the Clu+ regenerative program and a fetal gene expression signature marked by Sca1, but upon injury are unable to generate Sca-1+/Clu+ revSCs and could not regenerate the epithelium. This work demonstrates the role of TIPE0 in regulating the dynamic signaling that determines the injury response and enables intestinal epithelial cell regenerative plasticity.

[1] Department of Pathology and Laboratory Medicine, Perelman School of Medicine, University of Pennsylvania, Philadelphia, PA 19104, USA. [2] University of Pikeville—Kentucky School of Osteopathic Medicine, Pikeville, KY 41501, USA. [3] Centre for Systems Biology, Lunenfeld-Tanenbaum Research Institute, Mount Sinai Hospital, Toronto, ON, Canada. [4] Department of Biology, University of Maryland Baltimore College, Baltimore, MD 21250, USA. [5] Center for Applied Genomics, The Children's Hospital of Philadelphia, Philadelphia, PA 19104, USA. [6] Department of Pediatrics, Perelman School of Medicine, University of Pennsylvania, Philadelphia, PA 19104, USA. [7] Department of Molecular Genetics, University of Toronto, Toronto, ON, Canada. ✉email: goldsj@pennmedicine.upenn.edu; yhc@pennmedicine.upenn.edu

t was recently found that the stem cell niche and epithelium is regenerated after injury by de-differentiated adult cells[1], through a process that gives rise to Sca1+ fetal-like cells and is driven by a transient population of Clu+ revival stem cells (revSCs)[2–4]. The intestinal epithelial cells (IECs) that comprise the intestinal barrier protect the body from harmful luminal contents and play critical roles in nutrient absorption and waste excretion[5]. In a healthy gut, IECs are regenerated about every 5 days. During and after intestinal injury, such as through radiation, colitis, or ischemia, intestinal regeneration becomes more pronounced as the disrupted barrier is restored. Lgr5+/CBC (crypt basal columnar) stem cells were previously identified as responsible for daily homeostatic regeneration[6]; however, they are particularly susceptible to cell death during injury, and yet the intestine is still able to regenerate after injury. This has led to ongoing efforts to understand the molecular biology of injury-induced intestine regeneration[6].

Recently, it has been appreciated that all partially and fully differentiated epithelial cells studied thus far can revert to novel stem-cell states post injury and restore the homeostatic stem cell niche; a process named paligenosis[6,7]. A population of intestinal stem cells, known as "fetal-like" stem cells because of their similarity to fetal intestinal stem cells[8], has been identified in adult mice that are Sca-1+ and Lgr5−, and only appear in appreciable numbers after injury[3,4]. This fetal-like, Sca-1+ stem-cell program appears to require functional YAP/TAZ-signaling[9], which is known to be repressed by β-catenin signaling, the central pathway mediating intestinal stemness, regeneration, and differentiation[6]. Other recent work demonstrated that this regenerative program requires the transient induction of a small subset of Clu+ "revival stem cells" or rev-SCs (that are also Sca-1+), that are directly responsible for post-injury regeneration[2]. Thus, in the intestine, regeneration requires Sca-1+ and Clu+ cell states. However, the general signaling processes and critical proteins required to induce regenerative paligenosis after injury remain unknown.

The TNFAIP8-like (Tumor necrosis factor-alpha-induced protein 8-like, or TIPE) family of proteins (comprised of TNFAIP8 and TNFAIP8L1, TNFAIP8L2, and TNFAIP8L3) are homologous lipid transport proteins that help regulate PI3K-mediated signaling, and play crucial roles in inflammation, cell migration, and cell survival[10]. TNFAIP8L2, (aka TIPE2), the most well-studied member, is restricted to the immune compartment. Loss of TIPE2 results in defective leukocyte chemotaxis[11–13], and decreased dextran sodium sulfate (DSS)-induced colitis, with reduced immune cell migration to the gut[14]. In contrast, loss of TNFAIP8 (hereafter TIPE0), which is ubiquitously expressed[11,12], results in more severe DSS colitis; this effect was found to be independent of immune cells[15]. We report here that TIPE0 is a regulator of the intestinal injury response and controls intestinal cell stemness and plasticity during injury, by regulating basal Akt activation induced by the microbiota.

## Results

***Tipe0$^{-/-}$ mice are resistant to intestinal ischemia.*** To determine if TIPE0 modulated intestinal injury in an acute injury model without a regenerative component, we subjected both *Tipe0$^{-/-}$* and *Tipe2$^{-/-}$* mice to 60 min of distal ileal ischemia followed by 90 min of reperfusion (I/R90′). Compared with wild-type (WT) mice, *Tipe0$^{-/-}$* mice were resistant to intestinal injury, as were the *Tipe2$^{-/-}$* mice, but to a lesser degree Fig. 1a, b). The protection in the knockouts could not be explained by global differences in cytokine expression, which was induced to similar levels in all genotypes exposed to I/R90′ (Fig. 1c). Furthermore, we saw increased inflammatory cytokine expression in healthy

*Tipe2$^{-/-}$* mice as compared with WT controls, and increases in *Tnf* in the healthy *Tipe0$^{-/-}$* mice vs WT, all of which was in accordance with previous literature on these mice[10,14–17] and further suggested that decreased inflammatory signaling did not explain the protection seen. Bone marrow chimera studies using wild-type donors led to the induction of comparable I/R90′ injury in both the WT and TIPE2 mutant recipients, while TIPE0 mutant hosts still resisted tissue damage, like with the DSS model[14,15], suggesting that effects of TIPE2 or TIPE0 deficiency to I/R90′ injury are mediated by their specific roles in the immune or non-immune cells, respectively (Fig. 2a, b).

To determine whether the protection afforded by TIPE0-protein loss occurred early during the ischemia phase of injury, or during the reperfusion phase, where immune cell infiltration mediates further damage[18,19], mice were subjected to 60 min of ischemia without reperfusion (I60′). We found that *Tipe0$^{-/-}$*, but not *Tipe2$^{-/-}$*, mice were protected from I60′ injury (Fig. 2c, d), supporting a non-immune-mediated role for TIPE0 during ischemic injury. To further confirm that altered immune infiltration explained the findings in the *Tipe2$^{-/-}$* mice, we measured CD11b+ immune cell infiltration. After 60 min of ischemia, infiltration was expectedly minimal in WT and *Tipe2$^{-/-}$* mice, but became marked in only WT mice after a subsequent 90 min of reperfusion (Fig. 2e, f). The observed protection from ischemic injury correlated well with markers of intestinal cell death and apoptosis. Only *Tipe0$^{-/-}$* mice showed decreased levels of cell death via TUNEL staining, at both the I60′ and I/R90′ time points (Supplementary Fig. 1a, c) and activated caspase-3 staining at the I/R90′ time point (Supplementary Fig. 1b, c) matched the general (minimal) level of tissue injury observed. In healthy mice, we also saw no differences between WT and *Tipe0$^{-/-}$* mice in regard to activated caspase-3 staining but did a small increase in the *Tipe2$^{-/-}$* mice, perhaps due to the higher basal inflammation in these mice. To determine if this protection also extended to other forms of injury, *Tipe0$^{-/-}$* and *Tipe2$^{-/-}$* mice were subjected to radiation-induced intestinal injury. We found that only the *Tipe0$^{-/-}$* mice were found to be protected from injury, with no histological signs of post-injury regeneration 2 days after radiation (Supplementary Fig. 1d, e). Taken together, our results indicate that *Tipe0$^{-/-}$* mice are resistant to intestinal injury, which is immune-cell independent.

***Tipe0$^{-/-}$ epithelia have dysregulated growth and regeneration.*** To be able to further probe the role of TIPE0 in enterocytes, we recapitulated the injury resistance we saw ex vivo using enteroid culture. TNF signaling drives intestinal injury during ischemia therefore, TNF-mediated cell death was explored. 7-day (7d) old enteroids were exposed to 100 ng/mL TNF for 24 h and viability was assessed by Cell Titer Glo (CTG). *Tipe0$^{-/-}$* enteroids survived exposure to TNF much better than WT enteroids (90% vs 56%, Fig. 3a) and displayed enhanced survival for 24 h under hypoxic culture conditions (37% vs 24%, Fig. 3b). *Tipe0$^{-/-}$* were also better able to survive growth factor withdrawal, with greater survival after 1 day by CTG assay (88% vs 33%, Fig. 3c) and generally surviving one day longer than WT over the course of 5 days (Fig. 3c, d), as monitored by light microscopy. During these enteroid experiments, we noticed several other phenomena as well. Most strikingly, 1 day after growth factor withdrawal (by changing the culture media to DMEM/F12 media with 2 mM Glutamax from complete Intesticult™ murine enteroid media), new spheroids appeared in the ongoing *Tipe0$^{-/-}$* enteroid cultures, from places in each well where there was no visible structures or only small buds were growing previously (Fig. 2e, see below for further discussion). In addition, under regular culture conditions, we noticed marked differences in the *Tipe0$^{-/-}$*

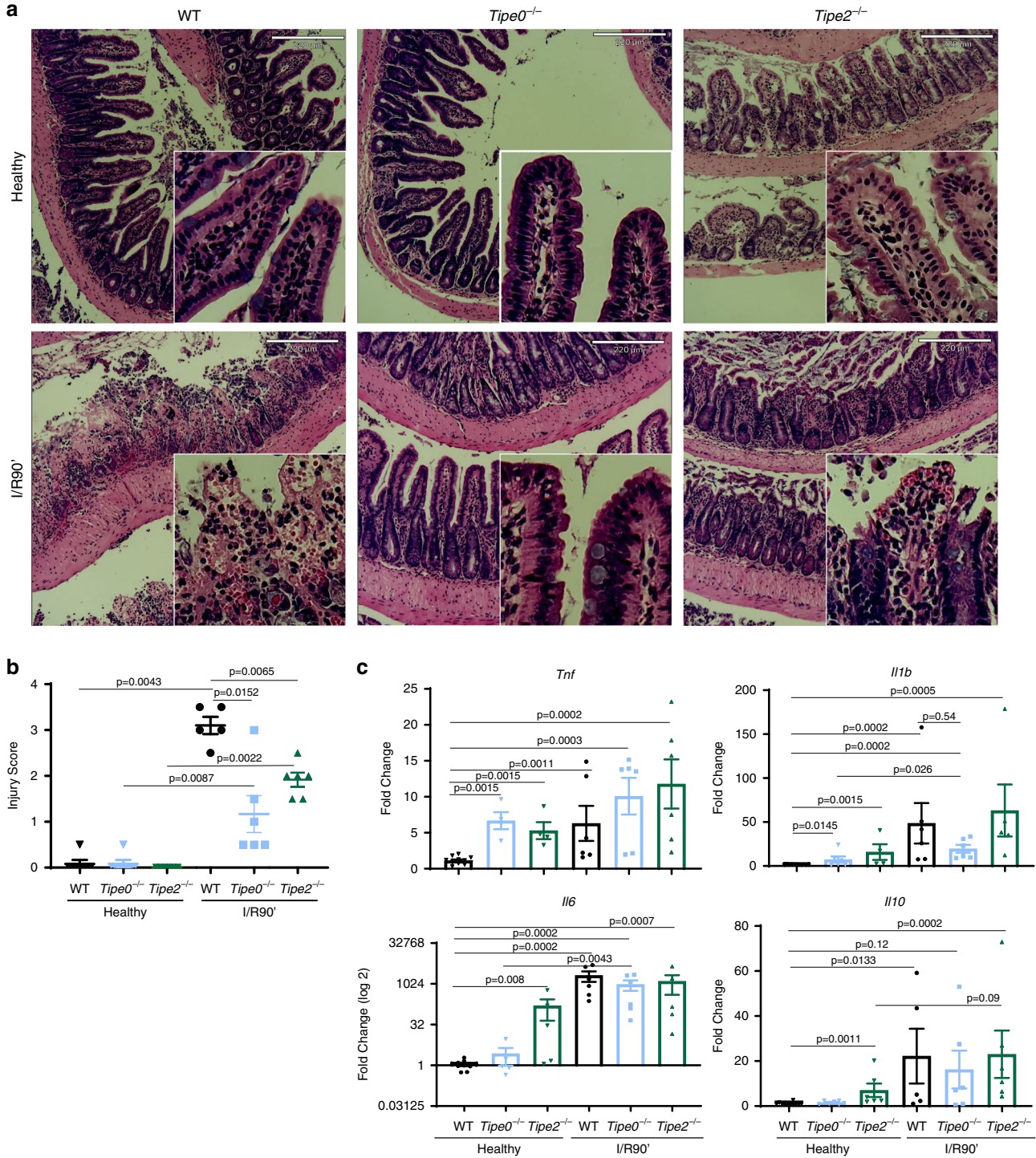

**Fig. 1 Loss of TIPE0 or TIPE2 results in protection from intestinal ischemia/reperfusion injury. a** Representative histological images and **b** Blinded histology scores in mice subjected to I/R90′, with adjacent healthy tissue as a control. $N = 5$ mice/group for WT & $Tipe0^{-/-}$ I/R90′; $N = 6$ mice/group for all other groups; bars = 220 μm; inserts are magnified 4×. Images are representative of the mean damage score for each condition. **c** RT-PCR, $N = 4$–7 mice/group, except for WT healthy $N = 11$ mice. For all panels $p$ is as indicated on the figure. Error bars are mean ± SEM. Multiple group comparisons were by Kruskal–Wallis one-way ANOVA. Two-tailed Mann–Whitney $U$ test was used to confirm ANOVA findings. Source data are provided as a source data file.

cultures in terms of proliferative capacity and enteroid size. $Tipe0^{-/-}$ enteroids appeared to be larger with altered morphology (Fig. 3f, g), and they were less plentiful than WT enteroids after 7 days of growth, even when controlling for crypt seeding density (Fig. 3h), suggesting that $Tipe0^{-/-}$ enteroids had

deficiencies in initiating post-isolation regeneration, but the $Tipe0^{-/-}$ mutant crypts that could grow were hyperproliferative. In addition, when doing crypt isolations using the same length of tissue, $Tipe0^{-/-}$ enteroids yielded less intact crypts that could be seeded (Fig. 3i).

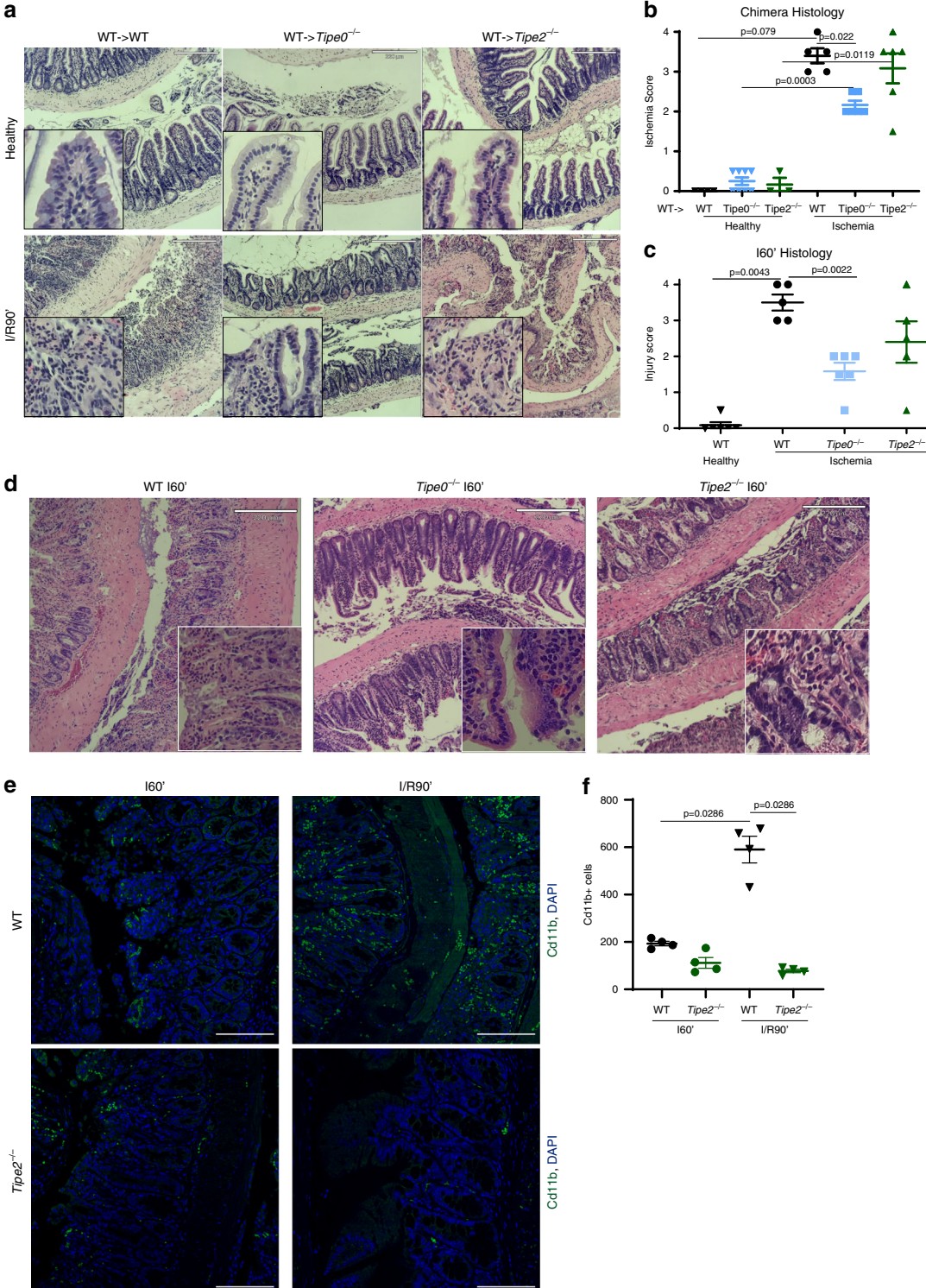

**Fig. 2 Protection from intestinal ischemia/reperfusion injury due to *Tipe0* gene deletion is immune-cell independent and occurs during the ischemic phase of injury in contrast to *Tipe2* gene deletion. a, b** Histology from bone marrow chimeras subjected to I/R90′, along with adjacent healthy tissue controls. Bars = 220 μm, inserts are 4× magnified, with blinded histology scores (**b**); Number of mice/group as follows: WT- > WT healthy = 4, WT- > *Tipe0⁻ᐟ⁻* healthy = 8, WT- > *Tipe2⁻ᐟ⁻* healthy = 3, WT- > WT ischemia = 5, WT- > *Tipe0⁻ᐟ⁻* ischemia = 6, WT- > *Tipe2⁻ᐟ⁻* ischemia = 6; images represent the mean histological score of each group. **c** Blinded histology scores and **d** representative images of mice subjected to 60 min of ischemia with no reperfusion time (I60′). N = 5 mice/group except WT Healthy and *Tipe0⁻ᐟ⁻* I/R60′, where N = 6; images represent the mean histological score, bars = 220 μm, inserts are 4× magnified. **e, f** CD11b staining (green) of WT and *Tipe2⁻ᐟ⁻* mice subjected to I60′ and I/R90′, image representative of 4/4 mice/group; bars = 100 μm; quantification in **f**. For all panels: *p* as indicated in the figure. For all graphs, error bars are mean ± SEM. Multiple group comparisons were by Kruskal–Wallis one-way ANOVA. Two-tailed Mann–Whitney *U* test was used to confirm ANOVA findings. Source data are provided as a source data file.

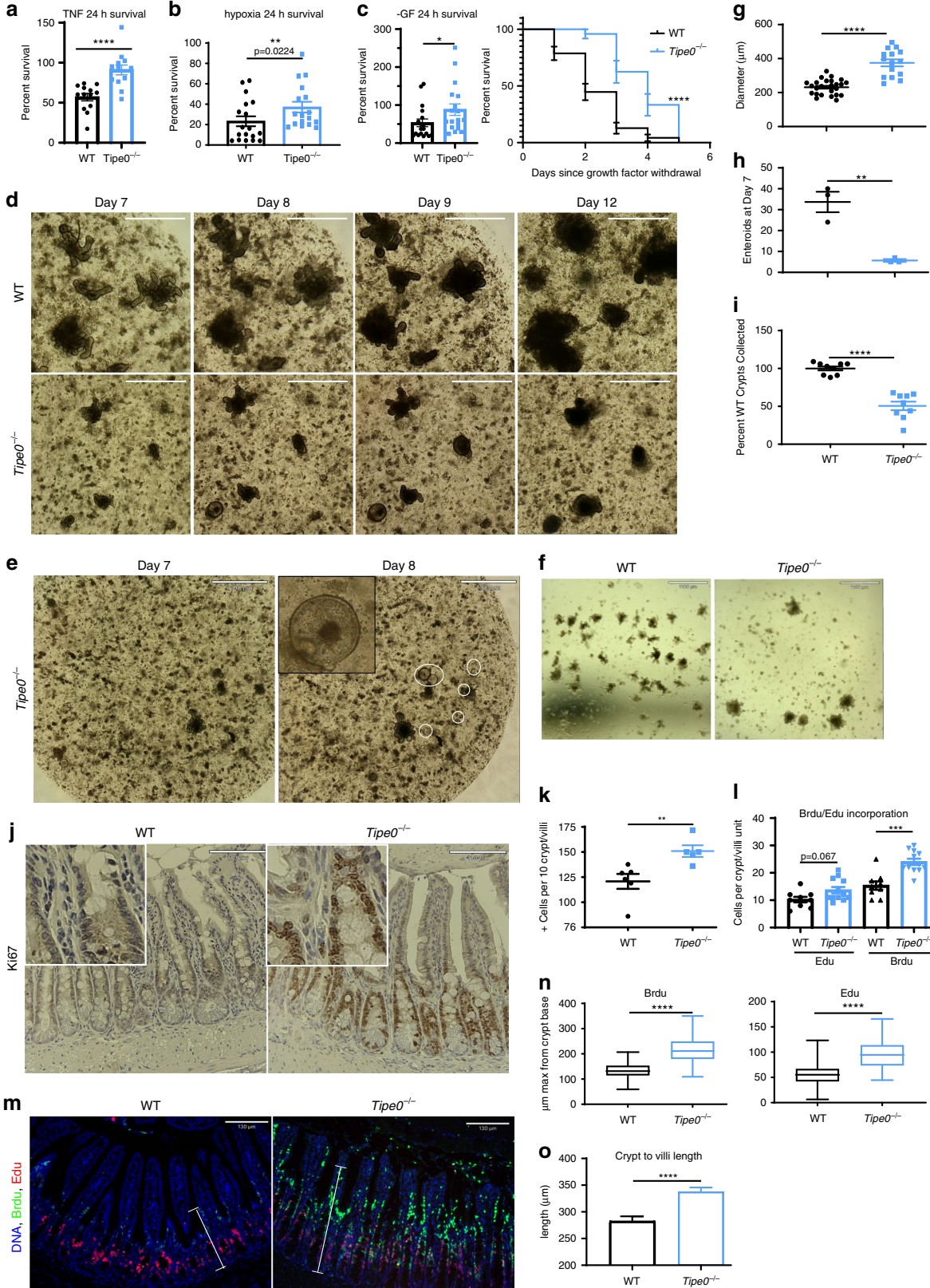

Given these findings we assessed proliferation in the native gut using several techniques. Ki-67 staining showed that more $Tipe0^{-/-}$ cells in the villi are proliferative (Fig. 3j, k). In addition, Brdu/Edu pulse chase assays further demonstrated that $Tipe0^{-/-}$ IECs migrate faster from the crypt to the tip of the villi than WT IECs and incorporated more nucleotide (Fig. 3l–n). We measured the crypt-to-villi-tip length and found this altered proliferation led to greater crypt-to-villi lengths in the knockout (Fig. 3o). Thus, $Tipe0^{-/-}$ enterocytes are resistant to various forms of injury in vitro, and their proliferating and regenerating ability were altered, with changes in downstream morphology.

**Fig. 3 $Tipe0^{-/-}$ enterocytes have dysregulated growth and responses to injury.** D7 enteroid survival (by CTG) after exposure to 24 h of 100 ng/mL TNF (**a**) or 24 h of hypoxia (**b** $p = 0.0224$); Total $N = 13$/group for **a**; total $N = 18$ for WT and 17 for $Tipe0^{-/-}$ for **b**; pooled from three independent experiments. **c** CTG survival assay 24 h after exchange into media lacking supplemental growth factors ($N = 18$/group, pooled from two independent experiments, $p = 0.0205$) with a 5-day survival curve by manual counting ($N = 47$ (WT), 23 ($Tipe0^{-/-}$); ****$p < 0.0001$ by Mantel–Cox test), with images (**d**) of the time course and development of de novo $Tipe0^{-/-}$ spheroids 1 day after growth factor withdrawal (**e**, white spheres). Bars = 1090 μm, image representative of three independent experiments, insert 5× magnified. **f** Day 7 enteroid cultures morphology. Bars = 1110 μm. Image representative of 5 independent experiments. **g** D7 enteroid diameters; $N = 29$ for WT, 30 for $Tipe0^{-/-}$. **h** Viable D7 enteroids after equal seeding, $N = 3$/group; $p = 0.0048$. **i** Viable ileal crypts; $N = 9$/group pooled across three independent experiments. **j** Ki-67 staining with quantification in **k**; staining from two independent experiments were pooled. Total $N = 6$ mice (WT), 5 ($Tipe0^{-/-}$); $p = 0.0087$ Images representative of the mean staining in each group. Bars = 110 μm. Inserts are 3× magnified. **l–o** Brdu/Edu pulse chase; $N = 3$ mice for WT, 4 for $Tipe0^{-/-}$ and staining was repeated twice. Quantification in **l**: three fields of view/mouse, total $N = 9$ (WT), 12 ($Tipe0^{-/-}$), $p = 0.067$ for Edu, $p = 0.005$ for Brdu. Images shown in **m** representative of staining in 3/4 mice in WT (Brdu failed to incorporate in one mouse), 4/4 mice in $Tipe0^{-/-}$; bars = 130 μm. Distance migrated by Edu+ and Brdu+ cells (white brackets), with quantification in **n**; $N > 60$ measurements/group and three random images per section analyzed. Box & whisker plots range from minima to maxima, with median as the center, and the bounds of the box from 25–75% percentile. O) Crypt-villi length from $N = 19$/group pooled from 4 mice/group. For all graphs, error bars are mean ± SEM, *$p < 0.05$; **$p < 0.01$; ***$p < 0.001$; ****$p < 0.0001$. Analyses done by two-tailed Mann–Whitney $U$ test unless otherwise indicated. Source data are provided as a source data file.

**$Tipe0^{-/-}$ intestines have altered differentiation**. Given the altered morphology and proliferation, we determined if intestinal differentiation was altered in the $Tipe0^{-/-}$ gut. We found less robust alkaline phosphatase in tissue (Fig. 4a), along with less intense, but more uniform Ker-20 staining in enteroids (Fig. 4b), indicating diminished terminal enterocyte differentiation. The knockout enteroids also lacked budding regions absent Ker-20 staining, indicating that regenerative programming was potentially disrupted (Fig. 4b, white dashed region in WT). We found increased mucus production and accumulation in goblet cells in the $Tipe0^{-/-}$ mice (Fig. 4c, d). In addition, there was less robust Paneth cell staining in $Tipe0^{-/-}$ tissue in the crypt bases (Fig. 4e, f), while in enteroids we saw clusters of Paneth cells (presumably indicating a stem cell niche) in locations outside the extending buds of the crypts, which suggested that some of the disorganization seen in $Tipe0^{-/-}$ enteroids could be due to abnormal stem cell loci (Fig. 4f, white circles).

Single-cell (sc)-RNASeq was employed to profile the intestinal epithelium in undamaged $Tipe0^{-/-}$ mice. Using previously established sc-RNASeq signatures we found that $Tipe0^{-/-}$ mice had equal or decreased levels of differentiated enterocytes, with a concomitant increase in a stem cell/partially differentiated group, Lgr5+ stem cells, and in secretory (goblet and enteroendocrine) cells (Fig. 4g, h; see Table S2 for the genes used to identify these groups). We also saw the same decrease in Paneth cells demonstrated in Fig. 4f. Together, these data suggest that $Tipe0^{-/-}$ intestines have disrupted differentiation with fewer terminally-differentiated enterocytes and more Lgr5+ stem cells and transitional epithelial cells.

**$Tipe0^{-/-}$ epithelia have dysregulation of the revSC program**. Recent work showed that acute injuries induce YAP signaling that activates the Sca-1+ (Ly6a) fetal-like program throughout the intestinal epithelium and transiently mobilizes Clu+ revSCs that are required to regenerate the intestine post-injury and form spheroids in enteroids culture systems The development of spheroids was reminiscent of what we saw in $Tipe0^{-/-}$ enteroids after growth factor withdrawal. While there were too few cells to identify the Clu+ rev-SC population in our undamaged sc-RNASeq data, we interestingly found decreased numbers of fetal-like stem cells in $Tipe0^{-/-}$ epithelia when compared with WT controls (Fig. 4h). However, when we performed confirmatory RT-PCR on the four principal genetic markers[8] of a fetal-like program (*Ly6a*, *Gja1*, *Spp1*, and *Tacstd2*), we surprisingly found that they were elevated in both isolated $Tipe0^{-/-}$ enterocytes and in healthy and ischemic $Tipe0^{-/-}$ ileal tissue (Fig. 5a, c), as well as

in irradiated tissue (Fig. 5e). In contrast, we saw appropriate induction of the fetal-like program[4] in WT and $Tipe2^{-/-}$ tissues upon irradiation, suggesting that dysregulation was specific to TIPE0 loss. In addition, previously described "+4 stem cell markers" were not uniformly elevated in either $Tipe0^{-/-}$ enterocytes or tissue samples (Fig. 5b, d). This split between a decreased number of fetal-like/Sca-1+ cells and increased expression of markers overall suggested that that TIPE0 is required for proper regulation of this program and that the program is dysregulated at baseline. Indeed, IF staining demonstrated that the unusual $Tipe0^{-/-}$ spheroids induced after growth factor withdrawal contained elevated number of Sca-1+ cells (Fig. 6a). In addition, we found that Sca-1 expression was enhanced in freshly isolated $Tipe0^{-/-}$ epithelia, when compared with WT controls, as measured by flow cytometry (Fig. 6b). Upon staining healthy ileum, we could also detect increased levels of epithelial Sca-1 (Fig. 6c, d; e-cadherin was used as an epithelial cell counterstain). Together, these data suggest that while Sca-1+ is being upregulated in the $Tipe0^{-/-}$ epithelium, the entire program does not properly engage to support regeneration.

We then performed bulk RNASeq analysis to evaluate if the post-injury regenerative program was inappropriately active at baseline, even if present in fewer cells. This revealed that the revSC markers were also aberrantly activated in $Tipe0^{-/-}$ enterocytes (Fig. 5f), similar to fetal-like program. Interestingly, we also found that enterocyte specific genes were upregulated in $Tipe0^{-/-}$ epithelia (Fig. 5g), despite our sc-RNASeq demonstrating they were less fully differentiated enterocytes, suggesting hyperactivation of the enterocyte program alongside the Sca-1+ and rev-SC programs, despite their decreased total numbers. Additionally, we observed increases in the goblet cell signature, in line with the increased numbers in the $Tipe0^{-/-}$ mice by staining and sc-RNASeq (Fig. 5g).

**$Tipe0^{-/-}$ mice cannot regenerate revSCs after DSS-injury**. $Tipe0^{-/-}$ mice have worse recovery and survival from DSS colitis[15], a model of injury that relies on the enterocytes to continually regenerate from injury, because most $Tipe0^{-/-}$ cells cannot survive the chemical insult. As mentioned previously, there was decreased enteroid generation with loss of $Tipe0$, implying that the $Tipe0^{-/-}$ DSS phenotype could be a result of deficits in regeneration initiation due to the dysregulation of the Clu+ regenerative program that is important in DSS colitis[2,9]. $Tipe0^{-/-}$ mice subjected to 7 days of DSS had a notable lack of Sca-1+ activation compared with strong induction of Sca-1+ cells observed in WT mice, which was absent in the $Tipe0^{-/-}$ mice (Fig. 6e–g). YAP-mediated signaling, which is upstream of the

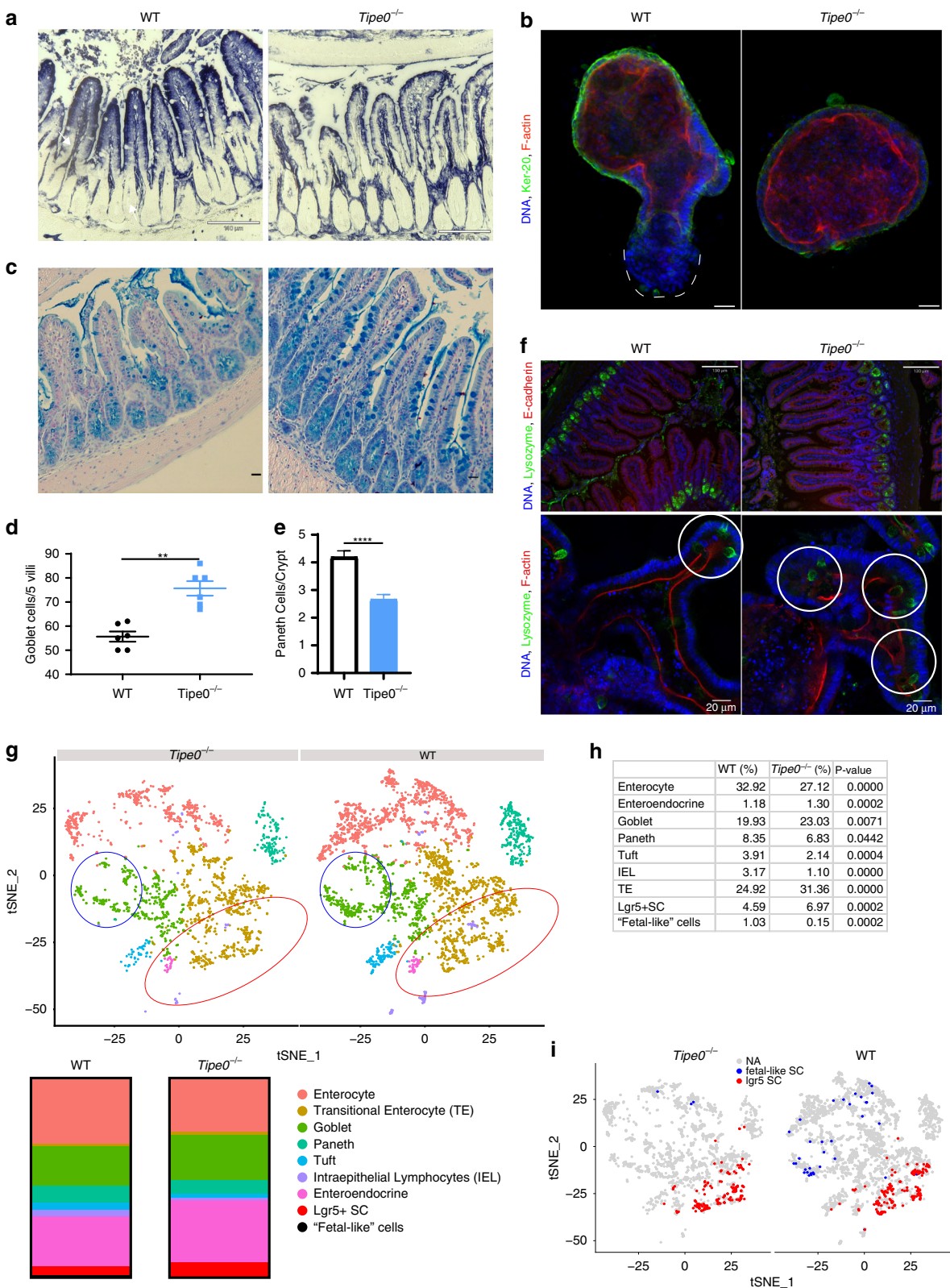

Figure legend:
Enterocyte
Transitional Enterocyte (TE)
Goblet
Paneth
Tuft
Intraepithelial Lymphocytes (IEL)
Enteroendocrine
Lgr5+ SC
"Fetal-like" cells

|  | WT (%) | Tipe0$^{-/-}$ (%) | P-value |
|---|---|---|---|
| Enterocyte | 32.92 | 27.12 | 0.0000 |
| Enteroendocrine | 1.18 | 1.30 | 0.0002 |
| Goblet | 19.93 | 23.03 | 0.0071 |
| Paneth | 8.35 | 6.83 | 0.0442 |
| Tuft | 3.91 | 2.14 | 0.0004 |
| IEL | 3.17 | 1.10 | 0.0000 |
| TE | 24.92 | 31.36 | 0.0000 |
| Lgr5+SC | 4.59 | 6.97 | 0.0002 |
| "Fetal-like" cells | 1.03 | 0.15 | 0.0002 |

Sca-1 response during DSS colitis[9] was also not induced in the enterocytes of the Tipe0$^{-/-}$ mice (Fig. 6h–j), and there was increased pYAPS127/YAP ratio, indicating YAP was more inactivated in the knockouts after colitis. We saw similar changes in both healthy ileum and colon at baseline by IF (Fig. 5h, i), and Western blot (Fig. 5j), with decreased nuclear YAP in Tipe0$^{-/-}$ mice and a concordant increase in the pYAPS127/YAP ratio.

Furthermore, during the regenerative phase of radiation-induced ileal injury and during DSS colitis, but not during acute ischemia, WT mice had increased levels of Tipe0 mRNA expression (Fig. 7a), lending credence to the theory that TIPE0 is important for maintaining signaling homeostasis during injury.

Asymmetric YAP activation is required for enteroid development[20], so we probed our Tipe0$^{-/-}$ enteroids to see if the

**Fig. 4 $Tipe0^{-/-}$ enterocytes have decreased differentiation markers and the intestine is shifted towards a less differentiated state.** All enteroid IF images representative of >3 d7 enteroids per genotypes, and three independent replicates were performed. **a** Alkaline phosphatase in intestinal tissue (purple, bars = 110 μm); images representative of mean staining seen in 4 WT and 5 $Tipe0^{-/-}$ mice. **b** Keratin-20 staining and F-actin staining in enteroids (IF, bars = 20 μm) with atypical symmetry in $Tipe0^{-/-}$ enteroids and dedifferentiated buds only seen in the WT (outlined with white dashed line). Images representative of >10 enteroids per group per experiment, with three independent experiments performed. **c** Alcian blue staining, bars = 20 μm, with quantification (**d** $p$ = 0.0022). Image representative of mean staining for 6 mice/group. **e, f** Paneth cell staining in ileal tissue and enteroids. **e** Paneth cells/ crypt ($N$ = 4 mice/group, 7 crypts/mouse, pooled) as determined by lysozyme staining in intestinal tissue (**f**); images representative of mean staining seen in each group. Staining also performed in enteroids (**f**, lower row). White circles indicate proliferative crypt foci as indicated by lysozyme staining in the enteroids. Bars = 130 μm for tissue, 20 μm for enteroids; counterstains (in red) as indicated. For enteroids, images representative of >4 enteroids per group in each of 3 independent experiments. **g–i** Single cell RNASeq of enterocytes, $N$ = 3 mice/genotype. WT = 3402 isolated cells, $Tipe0^{-/-}$ = 2006 isolated cells. Analysis of group populations by two-tailed $t$-test. **g** t-SNE plots by genetically defined cell populations with bar graphs and tabulated percentages of the major lineages (**h**). Blue circles indicate areas where fetal-like stem cells were found, while the red circles indicate the regions where $Lgr5^+$ stem cells were found. These specific, rare populations are highlighted separately in **i**. For all graphs, **$p$ < 0.01; $p$ < 0.001; ****$p$ < 0.0001 vs WT, or as indicated; error bars show mean ± SEM. Analyses done by two-tailed Mann–Whitney $U$ test unless otherwise specified. Source data are provided as a source data file.

regenerative deficits we saw in culture were associated with loss of YAP activation. Indeed, nuclear YAP, as determined by co-localization IF, was diminished in developing $Tipe0^{-/-}$ enteroids (Fig. 7b, c). Next, we probed for *Clu*, a selective marker of revSCs, by single-molecule in situ hybridization (sm-FISH; RNAScope), and found that like Sca-1, *Clu* was elevated in $Tipe0^{-/-}$ colons at baseline, but was poorly induced after DSS injury in the absence of TIPE0 (Fig. 7d, e).

**TIPE0 regulates microbiome-induced Akt activation**. To determine the mechanism for TIPE0 regulation of regenerative pathways, we undertook studies to identify which signaling pathways were regulated by TIPE0. Erk1/2 activation has been previously shown to modulate ischemia/reperfusion injury[21,22] but we saw no changes with loss of TIPE0 (Supplementary Fig. 2f). The TIPE proteins are global suppressors of PI3K-related signaling[10,23]. Given the central role of β-catenin in maintaining the stem cell niche[24] and our observed shift towards a de-differentiated state with an expansion of $Lgr5^+$ ISCs, as well as previous reports that YAP activity is inhibited by Akt[25] and β-catenin[6,26], we hypothesized that TIPE0 might regulate the PI3K/ Akt/β-catenin pathway. With loss of TIPE0, we found marked induction of pAktS473 and downstream β-catenin (including nuclear staining) at baseline in enteroid culture (Fig. 8a, b) and in freshly isolated enterocytes by immunoblotting (Fig. 8c). This pattern was also present in healthy and ischemic tissue by IHC (Figs. 8d, 9a–c, Supplementary Fig. 2g), and elevation in inter-mediary pGSK3βS9 was also seen (Fig. 9a, c). The levels of pAktS473 in healthy $Tipe0^{-/-}$ were much stronger in not just in enterocytes but also in the adjacent muscularis propria, We also saw elevated pAktS473 after radiation (Fig. 9d). Thus, TIPE0 is important for suppressing PI3K/Akt/β-catenin signaling.

Performing GSEA pathway analysis on our sc-RNASeq data, we were able to confirm that β-catenin pathways were elevated with loss of TIPE0 (Fig. 9e). Furthermore, genes known to have diminished expression with β-catenin knockout were elevated with TIPE0 loss (Fig. 9f) and conversely those genes that are elevated with β-catenin knockout were diminished (Fig. 9g), demonstrating that the high levels of β-catenin seen with $Tipe0^{-/-}$ were regulating downstream transcription of β-catenin-dependent genes. Ingenuity Pathway Analysis (IPA) was also performed on our sc-RNA-Seq data, and PI3K/Akt signaling was also found to be induced with loss of TIPE0 by this metric (Supplementary Fig. 3).

To determine if the basal Akt activation that TIPE0 inhibits was microbiome-dependent, we performed antibiotic-mediated microbiome ablation. After two weeks of broad spectrum antibiotics, we found the hyperactivation of Akt in $Tipe0^{-/-}$ mice to be completely lost in the gut (Fig. 8d, e). The increased β-catenin accumulation was also ablated by antibiotic administration (Fig. 9c, h). YAP is phosphorylated at S127 by Akt, and the increased pYAPS127/YAP ratio in the $Tipe0^{-/-}$ mice was also lost with antibiotic administration (Supplementary Fig. 4a–c). Since the commensal microbiome is required for the induction of basal intestinal PI3K activity[27], our results suggest that TIPE0 inhibits Akt activation through a PI3K-dependent mechanism.

**Injury resistance in $Tipe0^{-/-}$ mice is due to increased Akt and β-catenin activity**. To assess if the hyperactivation of Akt is a key mechanism by which loss of TIPE0 results in protection from acute intestinal injury, we administered the Akt-specific inhibitor GDC-0068[28] (which is currently undergoing Phase 2 trials) 2 h before ischemic injury and assessed the resulting injury. Akt-blockade completely ablated the protection conferred by loss of TIPE0 (Fig. 10a, b). To determine if downstream β-catenin accumulation was also important for the protection conferred by TIPE0 loss, we used our enteroid system due to the lack of specific in vivo β-catenin inhibitors. Administration of IWR-1-endo, a molecule that promotes Axin-mediated degradation of β-catenin, abolished the resistance to hypoxic death seen in $Tipe0^{-/-}$ enteroids (Fig. 10c). Using a similar approach, we found that non-Wnt-dependent β-catenin accumulation was responsible for development of the Sca-1$^+$ spheroids that appeared with growth factor withdrawal. Indeed, administration of Wnt antagonist IWP-2 did not block the formation of the Sca-1$^+$ spheroids, while administration of IWR-1-endo virtually abolished all Sca-1+ spheroids that appeared upon growth factor withdrawal (Fig. 10d).

To confirm the mechanism further, we used the CRISPR/ Cas9 system to ablate TIPE0 in the murine colorectal carcinoma cell line CMT-93. Using transiently-introduced Cas9, we were able to introduce insertions/deletions (indels) in the targeted TIPE0 region, with 80-90% efficiency (Supplementary Fig. 4d). We pooled our three clones that had the highest reduction in TIPE0 by indel and Western analysis and generated a polyclonal culture, as well a negative control (NC) polyclonal culture. By Western analysis, TIPE0 was reduced to 20% of the baseline, and so we deemed that these to be a TIPE0-knockdown line (TKD), but not a true knockout, likely due to the polypoidal nature of the CMT-93 cells (Supplementary Fig. 4e). With TIPE0-knockdown, we saw similar changes to those we saw previously in $Tipe0^{-/-}$ enterocytes, with increased pAktS473 and increased β-catenin levels (Supplementary Fig. 4e). Furthermore, we saw diminished growth in the CMT-93$^{TKD}$ cells, similar to the deficits seen in our $Tipe0^{-/-}$ enteroids (Supplementary Fig. 4f), without an accom-panying increase in cell death as measured by LDH (Supple-mentary Fig. 4g).

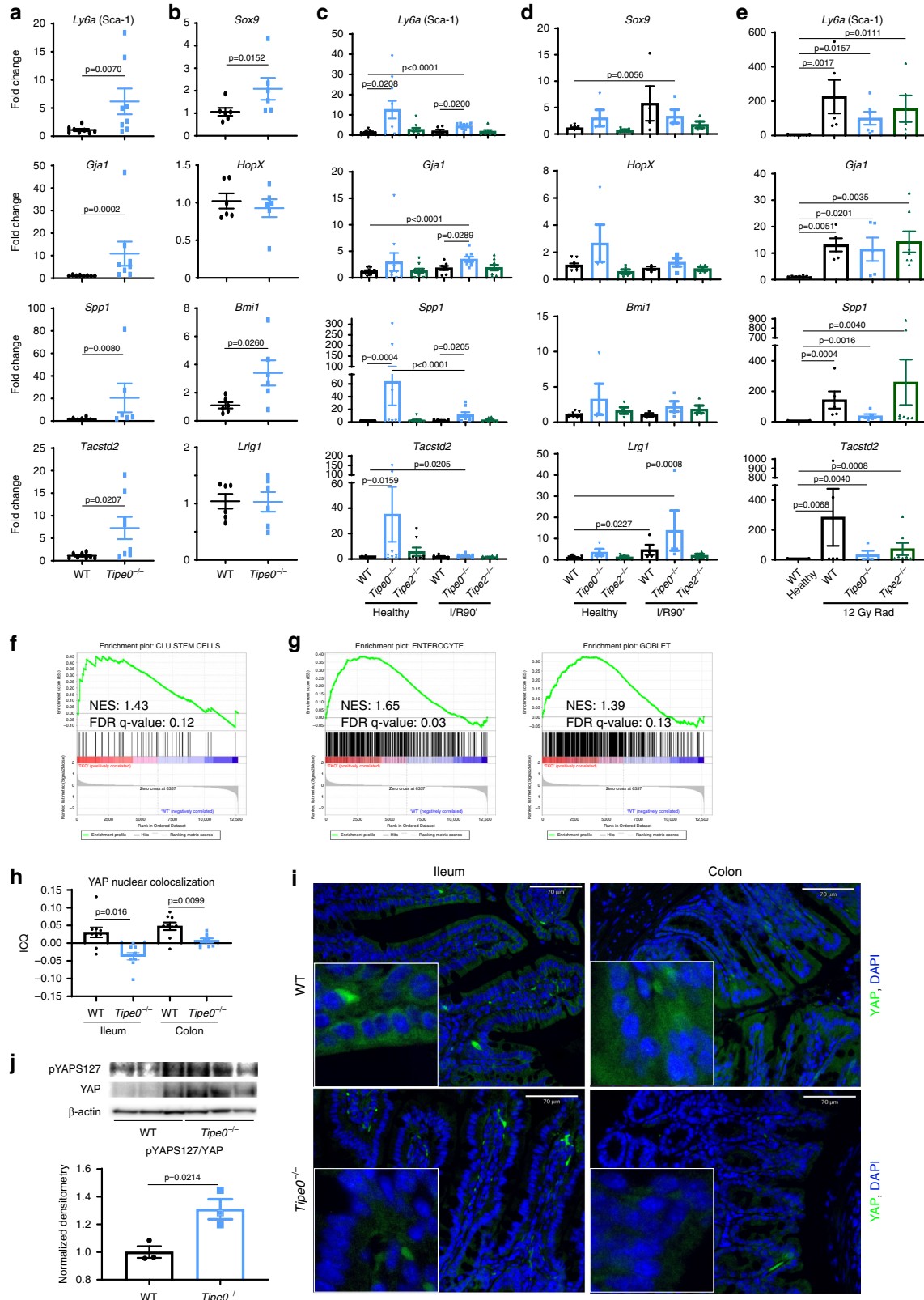

**TIPE0 extracts PIP2 inhibiting microbiota-induced signaling**. We previously reported that TIPE2 was able to extract PIP2 (Ptdlns(4,5)P$_2$) in the presence of PIP3 (Ptdlns(3,4,5)P$_3$), using a sedimentation-based assay[29]. To determine if TIPE0 could also do this, and in doing so modulate the levels of downstream PIP3 to effect the signaling changes we previously characterized, we used a new FRET-based extraction assay. With this system, we saw that TIPE0 could indeed extract PIP2, but only in the presence of PIP3 (Fig. 10e). In contrast TIPE0 did not extract PIP3 (negative values are indicative of binding without extraction that increases FRET efficiency).

**Fig. 5 Tipe0 gene deletion results in inappropriate basal activation of the Sca1+/Clu+ program that persists only if initial injury is resisted. a–e** RT-PCR in isolated ileal enterocytes (**a, b**) and ileal tissue (**c–e**). "Fetal-like," injury-response stem cell markers (**a, c, e**) and 4+ stem cell markers (**b,d**) analyzed by Kruskal–Wallis one-way ANOVA, with two-tailed Mann–Whitney $U$ test used to confirm ANOVA findings. **a** $N = 4$ mice/group with two replicates on different days pooled (total $N = 8$), except for $Tipe0^{-/-}$ Spp1, where $N = 6$ (from 3 mice x 2 replicates). **b** $N = 6$ mice/group. **c** $N = Y$ different-day-replicates from X total mice (Y rep, X mice) as follows: WT-healthy Ly6a, Gja $N = 18$ (4 rep, 5 mice); WT-healthy Spp1, Tacstd2 $N = 13$ (4 rep, 5 mice); for WT-I/R90' Ly6a, Tacstd2 $N = 6$ (2 rep, 3 mice); $Tipe0^{-/-}$ & $Tipe2^{-/-}$-healthy Ly6a $N = 10$ (2 rep, 5 mice); $Tipe0^{-/-}$ and $Tipe2^{-/-}$-healthy Gja1, $Tipe0^{-/-}$-healthy Spp1 $N = 9$ (2 rep, 5 mice); $Tipe0^{-/-}$-I/R90 all genes and $Tipe2^{-/-}$-I/R90' Gja1, Spp1 $N = 8$ (2 rep, 4 mice); $Tipe2^{-/-}$-I/R90' Ly6a, Tacstd2 and WT-I/R90' Spp1 $N = 7$ (2 rep, 4 mice); WT-I/R90' Ly6a, Tacstd2 $N = 6$ (2 rep, 3 mice). **d** $N = 4$ mice/group, except: $Tipe2^{-/-}$-healthy for Sox9, Bmi1, and Lrig1 ($N = 5$); WT-I/R90' for HopX and Bmi1 ($N = 3$). For WT-healthy, $N = 9$ (Sox9 and HopX) and $N = 12$ (Bmi1 and Lrig1), 5 mice/each with two replicates on different days pooled. (E) $N = 8$ mice/group for WT healthy, otherwise $N = 5$/group except: $Tipe0^{-/-}$ Ly6a $N = 6$, $Tipe0^{-/-}$ Tacstd2 $N = 4$, $Tipe2^{-/-}$ Gja1, Spp1 Tacstd2 $N = 7$. Bulk RNASeq performed on ileal enterocytes. GSEA data for the Clu+ Rev-SC program (**f**) and enterocyte lineage programs (**g**) presented with $Tipe0^{-/-}$ as positive correlation (red) and WT as a negative correlation (blue); $N = 3$ mice/group. YAP nuclear co-localization (**h**) by intensity correlation quotient (ICQ) in healthy WT and $Tipe0^{-/-}$ ileums and colons, with representative images of those quantified (**i**). $N = 3$ mice/group with three random images/mouse analyzed (total $N = 9$). Bars = 70 μm, inserts 4× magnified; **p < 0.01. **j** Western blot for pYAPS127 and total YAP in WT and $Tipe0^{-/-}$ ileal enterocytes. Densitometry normalized to WT with β-actin as the loading control. $N = 3$ samples/group, analyzed by two-tailed $t$-test. For all graphs error bars show mean ± SEM. Source data are provided as a source data file.

Using mass ELISA, we found significantly increased levels of ileal and colonic PIP3 in the $Tipe0^{-/-}$ mice, that was lost with antibiotic administration (Fig. 10f). As a confirmatory experiment, we then used co-localization analysis with a biosensor to look at membrane-associated PIP3 in our TKD CMT-93 cells. We electroporated a PIP3 (Akt-PH-EGFP) biosensor into our NC and TKD CMT-93 cells, and after 24 h of incubation, fixed the cells and measured the co-localization of the biosensor with a membrane-specific stain. With loss of TIPE0, we saw increases in PIP3 on the membrane (Fig. 10g, h) of the CMT-93 cells. Together, these results demonstrate that TIPE0 inhibits steady-state, microbiome-induced Akt activation in the gut through the basal extraction of PIP2, thereby reducing the amount of PIP3 that can be produced by commensal-microbiota-stimulated PI3K, and thus PIP3-dependent signaling through the Akt cascade.

## Discussion

The loss of TIPE0 fundamentally disrupts the ability of the intestine to appropriately engage in paligenosis after intestinal injury. With loss of TIPE0, IECs become insensitive to stimuli, ignoring usually injurious stimuli, but also lacking proper regenerative drive when injury does occur. This demonstrates that in a highly regenerative system such as the gut, regenerative capacity is often linked to earlier cell death, and that if cell death is disrupted, so too is the regenerative response.

We found that normally, TIPE0 inhibits commensal microbiome-induced Akt activation, and TIPE0 ablation leads to high levels of active Akt, which was directly responsible for the observed resistance to ischemic injury. Corresponding downstream increases in pGSK3βS9, increased β-catenin accumulation, and induction of β-catenin dependent genes were all also observed with the elevated pAktS473 levels, demonstrating activation of the entire cascade. This signaling pathway is important for cell survival during intestinal injury[30], and indeed pharmacologically mediated degradation of β-catenin negated the hypoxia resistance seen in $Tipe0^{-/-}$ enteroids, demonstrating that the entire axis is important for the protective effects seen with loss of TIPE0.

While the Wnt/β-catenin gradient is well-known to be central to intestinal differentiation and proliferation[1,6], the role of Wnt-independent Akt-modulated β-catenin in intestinal differentiation and plasticity is largely unknown. Interestingly, $Tipe0^{-/-}$ IECs appeared less fully differentiated by sc-RNA-Seq analysis and IF staining demonstrated disorganized differentiation in enteroid culture, although by GSEA these cells appeared to be hyperactive, with increased production of enterocyte genes, demonstrating high levels of Akt-mediated β-catenin signaling can alter intestinal differentiation. Furthermore, the $Tipe0^{-/-}$ enterocytes stayed in a more proliferative, less-differentiated state and had more rapid progression to the villus tip. Most strikingly, loss of TIPE0 disrupted post-injury regenerative plasticity; even though the $Tipe0^{-/-}$ gut had atypically high levels of the injury-response program at baseline, this pre-priming was deleterious, as there were less fetal-like cells overall, and no new Sca-1+ or Clu+ cells appeared after DSS injury to repair the intestinal barrier. $Tipe0^{-/-}$ crypts had similar deficits in their ability to form new crypts, despite the crypts that did form being larger. YAP has been shown to be critical for the both enterocyte development[20], and Sca-1+-mediated intestinal regeneration[9], and nuclear YAP was not properly induced in our budding $Tipe0^{-/-}$ crypts nor after DSS colitis. Akt activation has been shown to generally inhibit YAP activation through phosphorylation of S127[25] and Wnt-mediated β-catenin accumulation have been shown to inhibit YAP activity during intestinal regeneration by causing Lgr5+ ISC perdurance[26,31]. With loss of TIPE0 we saw this lack of YAP activation and increased pYAPS127/YAP ratio and increased numbers of Lgr5+ ISCs alongside β-catenin accumulation, and so we hypothesize that the dysregulation of Akt induced by loss of TIPE0 prevents the YAP activation needed for proper regeneration both directly and through β-catenin accumulation. Interestingly, Sca-1 itself can also be induced by β-catenin[32], and so the permanently-set high levels of active β-catenin and low levels of active YAP can also explain the dichotomy of the high Sca-1 levels we see at baseline, along with the lack of induction after injury.

The regulation of commensal microbiome-induced Akt appears fundamental to the ability for TIPE0 to enforce intestinal homeostasis, but relatively little is known about the mechanisms of TIPE0 other than it can bind phospholipids[33]. We found that TIPE0 is able to extract PtdIns(4,5)$P_2$ from the membrane, but only in the presence of PtdIns(3,4,5)$P_3$, indicating that this effect occurs when a cell is undergoing some form of PI3K-activating stimulation, such as by the commensal microbiome[27]. In doing so, TIPE0 restricts the total available PIP2 on the membrane, and thus limits the amount of PIP3 that can be induced, and indeed we saw significantly increased levels of PIP3 in Tipe0$^{-/-}$ ileums and colons that was lost with microbiota ablation. This is a previously undescribed mechanism regulating PI3K-dependent signaling, such as the Akt pathway. In the immune system, the closely related TIPE2 protein is responsible for enforcing a signaling concept known as local excitation, global inhibition (LEGI) that is critical for proper immune cell function[10,29,34]. Given that the $Tipe0^{-/-}$ gut is stuck in an inappropriate, hyperactive state with partial engagement of the fetal-like program, coupled with

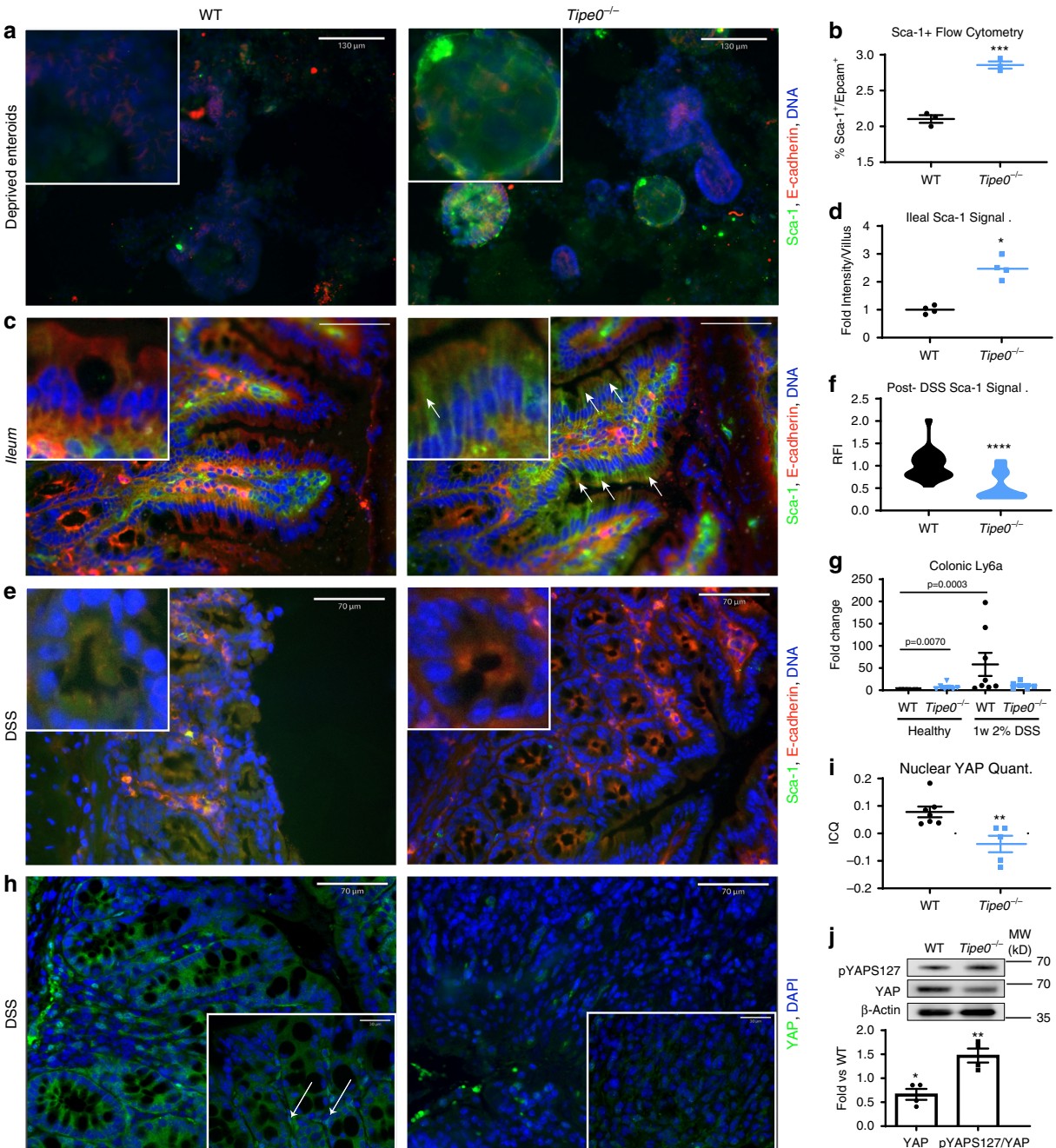

**Fig. 6 *Tipe0*<sup>−/−</sup> mice have inappropriate basal activation of the injury-response stem cell program, but an inability to induce the program to regenerate. a** Sca-1 staining in *Tipe0*<sup>−/−</sup> spheroids generated by depriving growth factor withdrawal; bars = 130 μm; insert 3× magnified. Representative of 3–4 *Tipe0*<sup>−/−</sup> spheroids/experiment, with three independent replicates. **b** Flow cytometry for Sca-1<sup>+</sup>/Epcam<sup>+</sup> cells. N = 3, representative of 2 independent experiments. p = 0.0005 two-tailed *t*-test with Welch's correction vs WT. **c** IF for Sca-1 and E-cadherin in healthy WT and *Tipe0*<sup>−/−</sup> tissue. E-cadherin was used as an enterocyte counterstain. Images representative of the mean intensity in 4 mice/group, with two fields of view/mouse randomly selected; bars = 130 μm; arrows mark enterocytes staining for Sca-1; insert 3× magnified. Sca-1 staining linearly enhanced in both images for easier merged channel visualization. **d** Quantification of enterocyte Sca-1 fluorescence intensity from **c**; N = 4 mice/group; p = 0.0286. **e** Sca-1 and E-cadherin staining from mice given DSS, with quantitation (**f**). E-cadherin is an used as an enterocyte counterstain. Quantification of total Sca-1 signal was by relative fluorescence intensity (RF), normalized to WT. N = 7 crypts/per mouse from 4 mice/group. Images representatives 3/4 mice evaluated for each group; bars = 70 μm; insert 3× magnified. **g** Colonic Sca-1 levels by RT-PCR. N = 7\(WT and *Tipe0*<sup>−/−</sup> healthy), N = 8 (WT DSS), N = 6 (WT and *Tipe0*<sup>−/−</sup>). Multiple group comparisons were by Kruskal–Wallis one-way ANOVA. Two-tailed Mann–Whitney *U* test was used to confirm ANOVA findings. **h** YAP staining in mice given DSS; bars = 70 μm, insert bars = 20 μm. Arrows show areas of high nuclear colocalization. Images representative of eight WT and five Tipe0<sup>−/−</sup> mice evaluated. **i** Quantification of nuclear YAP from **h** by Intensity Correlation Quotient (ICQ). N = 8 (WT), 5 (*Tipe0*<sup>−/−</sup>); p = 0.0025. **j** WB for pYAPS127 and YAP, with densitometry (N = 4/group); p = 0.0102 for YAP, p = 0.0021 for pYAPS127 by one-sample two-tailed *t*-test. For all graphs, *p < 0.05; **p < 0.01; p < 0.001; ****p < 0.0001 vs WT, or as indicated; error bars show mean ± SEM. Analyses done by two-tailed Mann–Whitney *U* test unless otherwise specified. Source data are provided as a source data file.

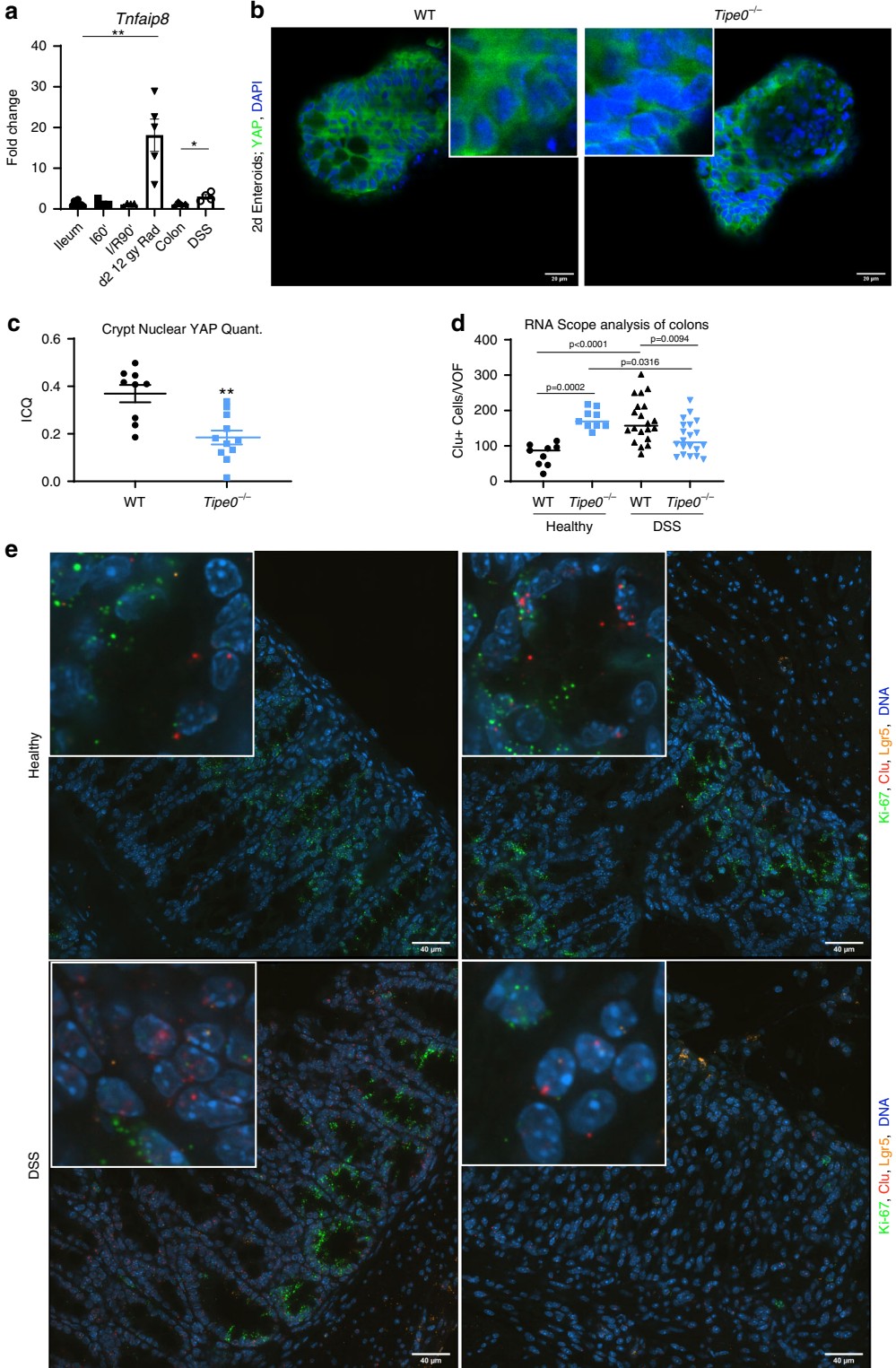

**Fig. 7 Nuclear YAP localization is inhibited in budding *Tipe0⁻/⁻* enteroids while Clu is induced in *Tipe0⁻/⁻* colons. a** RT-PCR assay for *Tnfaip8* (variant 1) gene expression. $N = 8$ for WT ileum, five for radiation, and four for all other groups; $p = 0.0016$ for radiation, $p = 0.0286$ for DSS. YAP staining in 2-day enteroids, bars = 20 μm; insert is 3× magnified, with quantification of nuclear YAP by ICQ in developing crypts (**c**). $N = 10$ for WT, $N = 12$ for *Tipe0⁻/⁻*, pooled from two independent experiments, with images representative of those quantified; $p = 0.0016$. **d, e** RNAscope Clu⁺ cells in mice WT and *Tipe0⁻/⁻* mice, before and after DSS treatment with additional staining for Ki-67, Clu, and Lgr5, with DAPI counterstain. Ki-67 and Lgr5 are used as controls to identify enterocyte specific staining for further quantification (**d**). For healthy tissues, $N = 9$/group (3 mice/group, with 3 images/group analyzed). For DSS tissues, $N = 20$ (five images each from $N = 4$ mice/group were analyzed). Images are representative of those quantified; Bars = 40 μm; insert is 5× magnified. **e** Data analyzed one-way ANOVA with Sidak's multiple comparison test. For all graphs, data analyzed by two-tailed Mann–Whitney *U* test unless otherwise specified; *$p < 0.05$; **$p < 0.01$ vs WT; error bars show mean ± SEM. Source data are provided as a source data file.

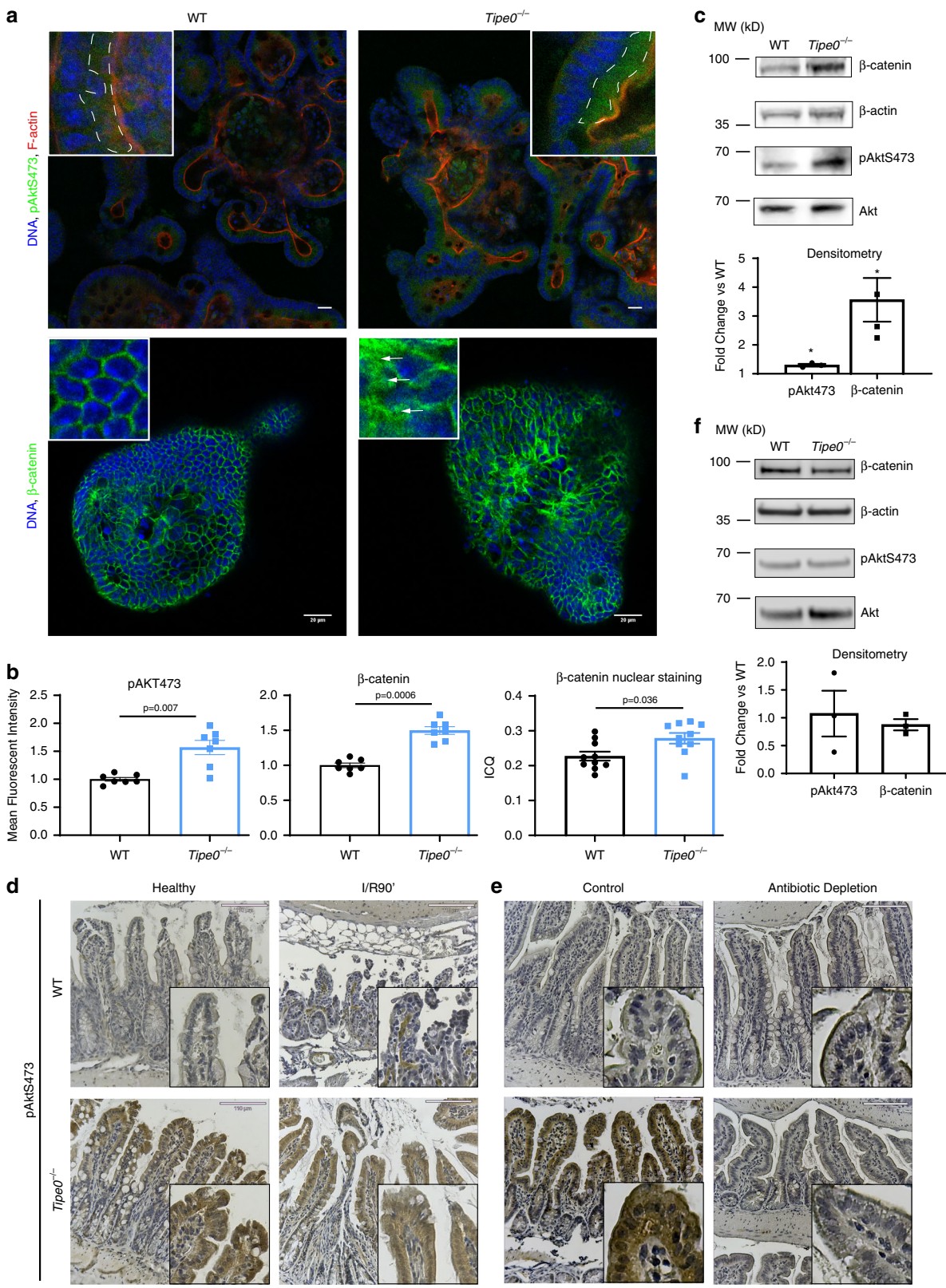

an inability to activate and regenerate after injury, it appears that LEGI is a signaling concept that is more universal than previously appreciated and its enforcement by TIPE0 appears to be vital for intestinal cell function. How the TIPE family in general induces LEGI, and specifically how TIPE0 does it, has unfortunately largely remained a mystery that we have partially solved here. This regulation of basal signaling phospholipid levels provides the mechanism for the global inhibitory effect of TIPE0. It remains to be determined how local excitation is enhanced by TIPE0, but it may be that in the absence of global inhibition, cells are simply less responsive to additional stimuli and thus TIPE0 works to enforce local excitation by enabling signaling flux.

**Fig. 8 Loss of Tipe0 results in microbiota-dependent hyperactivation of Akt signaling. a** IF for pAktS473 and β-catenin in enteroids. Counterstain as indicated. Bars = 20 μm; insert 4× magnified. Dashed lines encircle pAktS473 staining of interest; arrows indicate enhanced β-catenin staining. pAkt images are representative of seven enteroids/group across three independent experiments; β-catenin images are representative of ten enteroids/group across three independent experiments. **b** Quantification of pAkt and β-catenin staining from **a** as relative fluorescence intensity, with $N = 7$ enteroids/group pooled across three independent experiments. Nuclear β-catenin signal was also assessed by colocalization analysis (ICQ, intensity correlation quotient), $N = 10$ images/group pooled across three independent experiments. For all graphs, *$p < 0.05$; **$p < 0.01$; $p < 0.001$; ****$p < 0.0001$ vs WT, or as indicated; error bars show mean ± SEM. Analyses done by two-tailed Mann–Whitney $U$ test. **c** Western blot analysis for the indicated proteins in freshly isolated enterocytes, with densitometry analysis. Representative of three independent experiments for pAKT and four independent experiments for β-catenin. $p = 0.0199$ for pAkt, $p = 0.0429$ for β-catenin by one-sample two-tailed $t$-test) **d** Immunohistochemical staining for pAktS473 in healthy and ischemic (I/R90′) WT and $Tipe0^{-/-}$ mice. bars = 110 μm, inserts are twofold further magnified. Images representative of $N = 5$ mice per WT group (healthy and ischemic), or $N = 3$ for healthy $Tipe0^{-/-}$ mice and $N = 4$ for ischemic $Tipe0^{-/-}$ mice. Healthy samples were repeated 2–3 times as controls in the other experiments presented here, with consistent results. **e** IHC staining for pAktS473 in healthy WT and $Tipe0^{-/-}$ mice after 2 weeks of intestinal microbiota ablation with antibiotics. Images representative of 4 mice/group; bars = 110 μm. Control sections from mouse given house water are also shown for comparison. **f** Western blot pAktS473 and β-catenin in enterocytes from mice subjected to the same antibiotic treatment as in **e**, with densitometry analysis. $N = 3$ mice/group. Error bars indicate mean ± SEM (**b, c, f**). Source data are provided as a source data file.

This work demonstrates that TIPE0 is a regulator of the intestinal injury response and post-injury regeneration by serving as a regulator of the Akt/β-catenin signaling axis. TIPE0 serves as a central checkpoint in this signaling pathway, which we demonstrate is important for the regulation of plasticity that drives differentiation and de-differentiation. Without TIPE0, Akt signaling is inappropriately high, and IECs do not always fully differentiate, instead shedding into the lumen while still partially differentiated; additionally, they are unable to de-differentiate after injury to induce regeneration. This broad disruption in plasticity, and its resulting effects, have implications regarding intestinal wound healing as it relates to post-radiation/chemotherapy-induced injury, Inflammatory Bowel Disease, and colon cancer[7].

## Methods

**Animals**. Wild type (WT) C57BL/6 (B6) and CD45.1⁺ B6 mice were purchased from The Jackson Laboratory (Bar Harbor, ME) or bred in house. All mice used in this study were housed under pathogen-free conditions in the University of Pennsylvania Animal Care Facilities. All animal protocols used were pre-approved by the Institutional Animal Care and Use Committee of the University of Pennsylvania. $Tipe0^{-/-}$ and $Tipe2^{-/-}$ mice were previously generated, as described elsewhere[15,17].

**Bone marrow chimeras**. Bone marrow chimeras were generated as reported previously[14,15]. As a review, bone marrow cells were flushed from the femurs and tibias of donor mice (CD45.1). The red blood cells were lysed with ACK solution (8.29 g NH₄Cl, 1 g KHCO₃, 37.2 mg Na₂EDTA in 1 L of water). Cells were washed twice and re-suspended in cold PBS. Recipient mice (CD45.2) were sub-lethally irradiated with 500 rads twice separated by 4 h. The irradiated mice received a total of $10 \times 10^6$ donor bone marrow cells by tail vein injection 2 h after irradiation. 6–7 weeks after generation, peripheral blood leukocytes were collected and stained with CD45.1 and CD45.2 antibodies and measured by flow cytometry to determine the reconstitution rate. In the chimeric mice so generated, more than 90% of the hematopoietic cells were derived from donor bone marrow[15]. Mice were used 10-week post bone marrow transplant.

**Intestinal ischemia/reperfusion injury**. Distal ileal ischemia, with or without subsequent reperfusion, was induced as reported previously[30,35]. As a summary, mice were anesthetized under 1% isoflurane, supplemented by 10 mg/kg ketamine injected s.c. A midline laparotomy was made, and peripheral branches of superior mesenteric artery were occluded with aneurysm clips (Kent Scientific, Torrington, CT), to generate a 2- to 3-cm region of ischemic intestine adjacent to the cecum. Collateral blood flow through the intestine was blocked using aneurysm clips across the intestine and collateral vessels, demarking the region of ischemic intestine. Hematoxylin was administered to the edges of ischemic tissue to mark them, and then the incision was closed with surgical staples. Ischemia was maintained for 60 min, and then the incision was re-opened, the clamps were removed, and the incision was reclosed to induce reperfusion. The mice were maintained in a heated room for a variable amount of time (0, 1.5, or 4 h) with anesthesia for the reperfusion phase of injury. Equal numbers of male and female mice were used and pooled for all results.

Damage severity was evaluated using H&E-stained sections by blinded investigators. The scoring system is based on an IEC apoptosis/necrosis system,

where a score of 1 signified a loss of only the villus tips; 2, loss of 50% of the villus; 3, a loss of the entire villus, but with maintenance of the crypt; and 4, complete loss of the epithelial layer[30]. Fractional (to the nearest 0.5) scores were given, and the score was based on the average damage of the entire tissue section. Tissue samples were snap frozen in dry ice for subsequent RNA and protein analysis.

For the Akt-inhibition studies, mice were given 50 mg/kg GDC-0068 dissolved in standard MCT (methylcellulose Tween) vehicle (Sigma Millipore, Burlington, MA) by oral gavage 2 h prior to the initiation of surgery.

**Radiation-induced injury**. 8–12 week-old mice were subjected to 12 Gy whole body radiation and then sacrificed 2 days later. Damage severity was measured using a 0–4 scoring system, adapted from the I/R scoring system, but where crypt/villus damage was inverted (1 equals partial loss of crypts; 2 equals loss of crypts; 3 equals partial loss of villus; and 4 equals loss of the entire epithelial layer), in adaption of the known histopathology of this injury model[36,37]. Tissue samples were snap frozen in dry ice for subsequent RNA and protein analysis. Equal numbers of male and female mice were used and pooled for all results.

**DSS-induced injury**. To induce experimental colitis, mice were administered to Affymetrix brand 2.0% dextran sodium sulfate (DSS) (Thermo Fisher Scientific, Waltham, MA) drinking water for 7 days, followed by immediate sacrifice. Weight, presence of bloody stool, and diarrhea were monitored to track the progression of colitis. Damage severity was evaluated using H&E-stained sections by blinded investigators using a scoring system from 0–40 that accounts for epithelial damage and immune infiltration[38]. Distal colonic colon samples were snap frozen in dry ice for subsequent RNA and protein analysis. Only female mice were used for these studies.

**Microbial depletion**. Mice were given a cocktail of 1 g/L metronidazole (Santa Cruz Biotechnology, Santa Cruz, CA), 1 g/L neomycin (Henry Schein Animal Health, Dublin, OH), and 0.5 g/L vancomycin (Sigma Millipore) ad libitum for 2 weeks in their drinking water prior to sacrifice and tissue collection. Equal numbers of male and female mice were used and pooled for all results.

**Ileal enteroid cultures**. Several centimeters small intestine was dissected out of the mouse, briefly vortexed in cold Dulbecco's Modified Eagle Medium (DMEM), transferred to 10 mL Reagent 1 (10 mL CMF-HBSS and 20 μl 0.5 M NAC) on ice, and was vortexed every 15 s for two min. Tissue was then transferred to 10 mL cold Reagent 2 (10 mL CMF-HBSS + 1 mM NAC and 200ul 0.5 M EDTA). Tube was then placed onto gentle rotor for 45 min at 4 °C, while 50 ml conical vial was coated with 1% bovine serum albumin (BSA) on rollers at 4 °C. Conicals of tissue were placed on ice and vortexed for 30 s, 3 times. Solution was poured over 70 μm BD filter, placed into a BSA-coated 50 ml conical. Eluate was centrifuged at $300 \times g$ for 3 min and resuspended in Basal media (advanced DMEM/F12, 1X Glutamax, 1X HEPES, 1X Penn/strep) on ice. Crypt concentration was then measured and tissue culture plate was warmed and moistened with 1X PBS in incubator (37 °C). Crypts were pelleted in mini-fuge for 30 s and equal concentrations of crypts were resuspended in 80% matrigel/20% Intesticult media (Stem Cell, Cambridge, MA). PBS was removed from the warmed plate and crypt/matrigel suspension was pipetted into the middle of the well. Matrigel was then allowed to polymerize for 10 min at 37 °C and Intesticult media plus antibiotic–antimycotic (Gibco, Gaithersburg, MD) was added to wells (400 μl/well for 4-well chamber-slides, 100 μl/well for 96-well plates) and changed every 2–3 days. Enteroids were always collected from gender-matched pairs (a combination of male and female mice were used for various independent replicates, and were randomly selected). Mice were also age matched, and were 6–12 weeks of age.

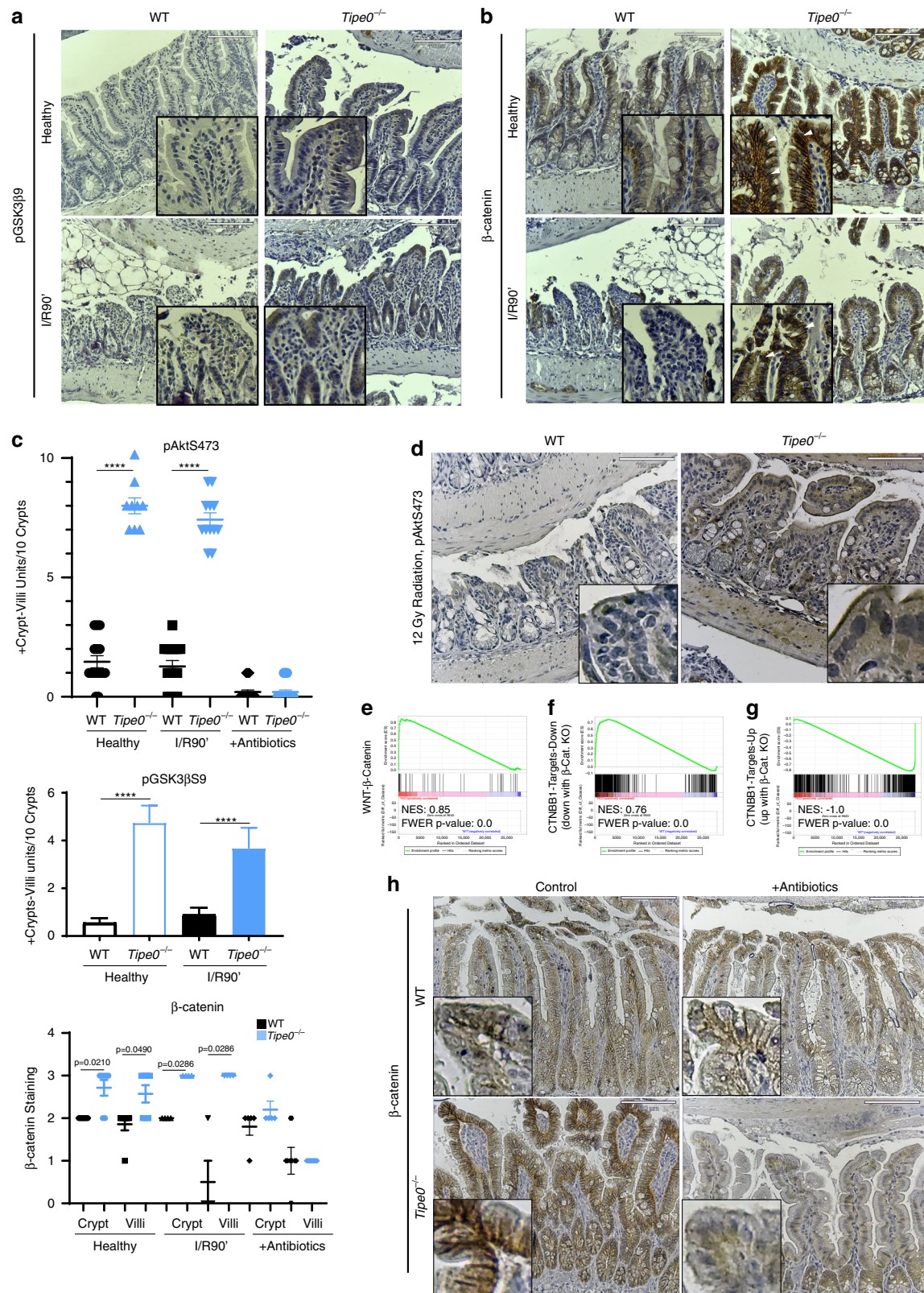

**Sample collection for frozen section slides**. The colon and/or small intestine was dissected and flushed with ice-cold PBS, longitudinally splayed, swiss rolled, snap frozen in a isopentane and dry ice mixture, embedded in OCT compound, and then sectioned with a cryostat. Tissue was then mounted on gelatin-coated histological slides and stored at −70 °C.

**Sample collection for paraffin-embedded slides**. Mice were anesthetized using isoflurane, and then sacrificed by cervical dislocation. The intestine was dissected

and flushed with ice-cold PBS, longitudinally splayed, Swiss-rolled, fixed in 10% formalin for 24 h, stored in 70% EtOH, and then embedded in paraffin.

**RNA isolation and real-time PCR**. RNA was isolated using TRIzol (Thermo Fisher), followed by a Qiagen RNeasy kit, per manufacturer protocol (Qiagen, Valencia, CA) and reverse transcribed to cDNA with a high capacity reverse transcription kit (Applied Biosystems, Foster City, CA) Real-time PCR was performed using an ABI 7500 Fast Real-Time PCR System (Applied Biosystems) or a

**Fig. 9 The PI3K/Akt/β-catenin signaling axis is upregulated with TIPE0-ablation.** Immunohistochemical staining for **a** pGSK3βS9 and **b** β-catenin in healthy and ischemic (I/R90′) WT, and $Tipe0^{-/-}$ mice. pGSK3βS9 images representative of 7/7 mice in each WT group (healthy and ischemic) and 5/5 mice in each $Tipe0^{-/-}$ group (healthy and ischemic). β-catenin images representative of four mice per group for ischemic images and seven for the healthy images. Bars = 110 μm, inserts are twofold further magnified. Arrowheads highlight enhanced nuclear β-catenin staining. **c** Quantification of histology from Figs. 8d, e, and 9a, c, h. For pAKT and pGSK3β staining, three random sections/slide were analyzed and crypt-villi units with more than 50% of enterocytes staining positively per ten units were quantified, except for the pGSK3β staining, for which one WT-H and one WT-I/R90′ did not stain. Total N for pAKT: 15 (WT-H and WT-I/R90′) 9 ($Tipe0^{-/-}$-H), 12 ($Tipe0^{-/-}$-I/R90′), 25 (antibiotic samples). Total N for pGSK3β = 15 ($Tipe0^{-/-}$ samples) and 20 (WT samples). For β-catenin staining a score from 0 to 3 was assigned to assess the strength of staining. N = 7 for healthy tissues, N = 4 for ischemia tissues, N = 5 for tissues from mice given antibiotics. N for all quantification is the same as for their respective images. **d** IHC for pAktS473 in mice 2 days post 12 Gy radiation, bars = 110 μm, Images representative of 4 mice/group; inserts are 4× magnified. **e–g** GSEA of Sc-RNASeq data from isolated enterocytes. TKO ($Tipe0^{-/-}$ mice) are positively correlated in all images. N = 3 mice/group, pooled prior to sequencing. **e** Wnt/β-catenin signaling axis; **f** genes that are downregulated with loss of β-catenin (thus are upregulated by β-catenin); **g** genes that upregulated with loss of β-catenin (thus are downregulated by β-catenin). **h** β-catenin IHC in healthy mice with and without antibiotics to ablate the microbiota. Images representative of 4 mice per group. Bars = 110 μm; inserts 4× magnified. For all graphs error bars show mean ± SEM; and analyses were by two-tailed Mann–Whitney U test; ****p < 0.0001. Source data are provided as a source data file.

Quantiflex 12X Real-Time PCR system (Thermo Fisher). Relative fold changes were determined using the ΔΔC$_T$ calculation method. Values were normalized to the internal control, 18 s ribosomal RNA, that was amplified using QuantiTect Primers (Qiagen). Primers all of which were manufactured by Integrated DNA Technologies (Coralville, Iowa) are as follows: *Il6* (5′-CGGAGGCTTAATTACA CATGTT-3′ and 5′-CTGGCTTTGTCTTTCTTGTTATC-3′), *Il1b* (5′-GCCCATC CTCTGTGACTCAT-3′ and 5′-AGGCCACAGGTATTTTGTCG-3′), *TNF* (5′-AT GAGCACAGAAAGCATGATC-3′ and 5′- TACAGGCTTGTCACTCGAATT-3′), *Il10* (5′-ATTTGAATTCCCTGGGTGAGAAG-3′ and 5′-CACAGGGGAGAAATC GATGACA-3′), *Ly6a (Sca-1)* (5′-GAAAGAGCTCAGGGACTGGAGTGTT-3′ and 5′-TTAGGAGGGCAGATGGGTAAGCAA-3′), *Hopx* (5′-CGGAGGACCAGGT GGAGAT-3′ and 5′-CCGGGTGCTTGTTGACCTT-3′), *Gja1 (Cnx43)* (5′-TGGG GGAAAGGCGTGAGGGA-3′ and 5′-ACCCATGTCTGGGCACCTCTCTT-3′), *Sox9* (5′-CAAGACTCTGGGCAAGCTC-3′ and 5′-GGGCTGGTACTTGTAAT CGG-3′), *Bmi1* (5′-CTGGAGAAGAAATGGCCCACTA-3′ and 5′-CTCATCTT CATTCTTTTGCAAGTTG-3′), *Spp1* (5′-TCGGAGGAAACCAGCCAAGGAC T-3′ and 5′-AAGCTTCTTCTCCTCTGAGCTGCCA-3′), *Lrig1* (5′-GAACACCTG AACCTTGGAG-3′ and 5′-CTGCAGCATCCTACCCATTAG-3′), *Tacstd2 (Trop2)* (5′-GAACGCGTCGCAGAAGGGC-3′ and 5′-CGGCGGGCCCATGAACAGTG A-3′). Axin2 was measured using QuantiTec Primers (Qiagen).

**Frozen section immunofluorescence.** Frozen sections were washed in Formaldehyde Fixative Solution (85 mM Na$_2$HPO$_4$, 75 mM KH$_2$PO$_4$, 4% paraformaldehyde, and 14% (v/v) saturated picric acid, pH 6.9) for 20 min at −20 °C, upon removal from the freezer. Following fixation, slides were washed 3 × 5 min in TBS, blocked in a solution of 150 ul normal goat serum/10 mL TBS for 1 h, and then administered primary antibody diluted in blocking solution with overnight incubation at 4°C. Primary antibodies and dilutions were as follows: Normal Rabbit IgG, used as IgG control at same concentration as Primary antibody (Cell Signaling, Danvers, MA); Sca-1 rat anti-mouse E13-161.7 at 1:50 (122502, Biolegend, San Diego, CA); anti-E-cadherin mouse mAb at 1:50 (14472, Cell Signaling); anti β-catenin rabbit mAB at 1:100 (8480, Cell Signaling).

Slides were washed with TBS. Secondary antibody was added to slides for 1 h, protected from light. Slides were washed and mounted with VECTASHIELD Hardset Antifade Mounting Medium with DAPI (Vector Laboratories, Burlingame, CA). Secondary antibodies and dilutions were as follows: Anti-mouse IgG (H + L), F(ab′)2 Fragment (Alexa Fluor®555 Conjugate) at 1:500 (#4409, Cell Signaling Technology); Anti-rabbit IgG (H + L), F(ab′)2 Fragment (Alexa Fluor®488 Conjugate) at 1:500 (#4412, Cell Signaling Technology); Anti-rab IgG (H + L), F (ab′)2 Fragment Alexa Fluor®488 Conjugate) at 1:500 (#4416, Cell Signaling Technology).

**Paraffin-embedded IHC and IF.** Paraffin-embedded tissue sections were prepared according to the manufacturer's protocol for the corresponding primary antibody, and primary antibody concentrations were as per the manufacturer's datasheet. An appropriate Vectakit Elite ABC (anti-rabbit or anti-mouse, Vector Labs) was then used per manufacturer's protocol. DAB reagent was from Dako (Agilent, Santa Clara, CA). Slides were then counterstained with Hematoxylin (Gill No. 3 strength, Thermo Fisher) For IF, primary and secondary antibodies, dilutions, and incubation times were the same as for frozen sections, with additional antibodies listed below. Slides were washed and mounted with VECTASHIELD Hardset Antifade Mounting Medium with DAPI (Vector Laboratories).

Primary IHC antibodies were: anti-β-catenin rabbit mAb (8480, Cell Signaling), anti-pGSK3βS9 rabbit mAb (9323, Cell Signaling); anti-pAKTS473 rabbit mAb (4060, Cell Signaling); anti-activated-Caspase-3 (AF835, R&D Biosystems, Minneapolis, MN); anti-Ki67 (ab16667, Abcam, Cambridge, England); anti-pERK1/2 (4370, Cell Signaling).

Additional primary IF antibodies and dilutions were: anti-CD11b at 1:200 (NB110-89474, Novus Biologicals, Centennial, CO); anti-human lysozyme EC 3.2.1.17 at 1:200 (Agilent); anti-YAP rabbit mAb at 1:100 (14074, Cell Signaling).

Alkaline Phosphatase staining was performed with a BCIP/NBT substrate kit (Vector Laboratories) per manufacturer's protocol.

During quantification, naïve tissue and healthy adjacent tissue were both quantified, and if no differences were found between these two groups within a genotype, these groups were pooled for the pGSK3βS9 and β-catenin staining.

**Enteroid/enteroid IF.** Enteroid staining was done following a previously published protocol[39]. Primary and secondary antibodies were used at the same dilutions as listed above. Additional primary antibodies and stains are: anti-human Keratin-20 at 1:400 (13063, Cell Signaling); phalloidin-555 at 1:100 (8953, Cell Signaling), anti YAP at 1:100 (14074, Cell Signaling). Alkaline phosphatase staining was performed using the BCIP/NBT substrate kit (Vector Laboratories) following a published protocol[39].

**TUNEL assay.** Fluorometric TUNEL assay was performed using a DeadEnd Kit (Promega, Madison, WI), in accordance with the manufacturer's protocols.

**Brdu/Edu pulse chase experiment.** Mice were injected i.p with 50 mg/kg BrdU followed by 5 mg/kg Edu 22 h later. Mice were then sacrificed 2 days after the initial injection. Biotin anti-Brdu MoBU-1 at 1:25 (317904, Biolegend) and Alexa Fluor®488 streptavidin at 1:200 (016-540-984, West Grove, PA) was used to detect Brdu while Alexa a Fluro®555 Click-iT Edu (Thermo Fisher) kit was used to visualize the Edu; the combined staining protocol was done per the manufacturer's protocols.

**Growth factor withdrawal assays.** Day 7 WT and TIPE0$^{-/-}$ enteroids were deprived of all growth factors for several days by exchanging Intesticult media for basal DMEM/F12 media (as described above) and monitored by microscopy. Relevant signaling was analyzed with co-culture with either 100 μM IWP-2 (sc-252928A, Santa Cruz Biotechnology, Dallas, TX) or 10 μM IWR-1-endo (sc-295215A, Santa Cruz Biotechnology, Dallas, TX) that was added at the time of growth factor withdrawal.

**Enteroid viability and growth assays.** Two identical 96-well plates were seeded with WT and TIPE0$^{-/-}$ enteroids. On Day 7 one plate was placed in a hypoxic chamber for 24 h (1% O2, 5% CO2, 94% N2), with or without addition of 10 μM IWR-1-endo to some well. The other plate was kept in standard culture conditions, also with or without addition of IWR-1-endo. For TNF-mediated death, 100 ng/mL TNF was added to media for 2 days, in parallel to a control plate given just vehicle. A CellTiter-Glo® (CTG) Luminescent Cell Viability Assay (Promega, Madison, WI) was then performed in accordance with the manufacturer's protocols. Luminescence was measured on the Tecan Infinite M200 PRO (Tecan, Morrisville, NC). Injury conditions were normalized to the non-injured control plate.

**Western blot.** Briefly, 40 μg of total cell lysates were run on 4-12% gradient SDS-PAGE gels. Blocked nitrocellulose membranes were probed overnight with primary antibody in LiCOR TBS Blocking Buffer/0.1%-Tween solution, rocking at 4 °C. Primary antibodies used were the same as above, at manufacturer recommended dilutions, with addition of anti-mouse β-actin (3700, Cell Signaling) and anti-pYAPS126 (13008, Cell Signaling). After incubation, membranes were washed with 0.2% TBS-Tween solution and incubated with the appropriate HRP-conjugated secondary antibody at 1:3000 dilution (GE Healthcare, Marlborough, MA). After

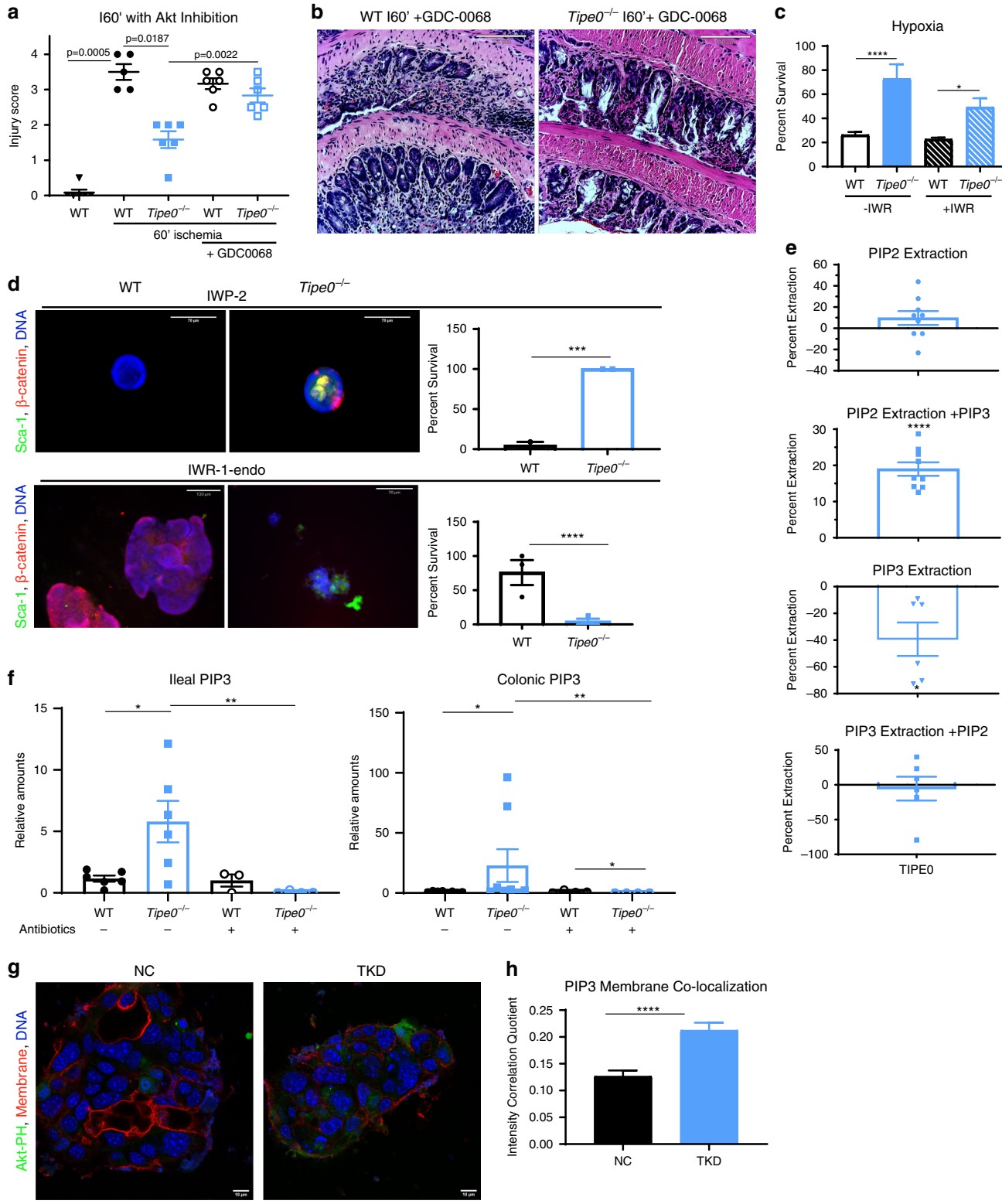

final washes with 0.2% TBS-Tween and 1X TBS, membranes were analyzed with Pico or Femto ECL reagent (Thermo Fisher) using a LiCor Odyssey imaging system (Li-cor, Lincoln, NE). For the β-catenin Western blot in Fig. 8C, Jackson ImmunoResearch (West Grove) 800 nm anti-Rabbit was used as a secondary and imaged directly with the LiCor system. All cropped blot images in the final figures are outlined in black and all compared bands were processed in Adobe Photoshop equally and at the same time using the Auto Tone function for image clarity. Raw images were used for densitometry analysis. Uncropped blots are provided in the source data file.

**RNAScope**. smFISH (RNAScope) was performed according to the manufacturer's recommendations, using their pre-made probes.

**Bulk RNASeq**. Enteroids were isolated as described above. RNA-Seq was performed by BGI (Shenzhen, China). Total RNAs were extracted from samples, then mRNA and non-coding RNAs were enriched by removing rRNA from the total RNA with kit. By using the fragmentation buffer, the mRNAs were fragmented into short fragments (about 200–500 nucleotides (nt)), then the first-strand cDNA was

**Fig. 10 The increased Akt signaling in $Tipe0^{-/-}$ mice is responsible for injury resistance and is due to loss of TIPE0-mediated regulation of basal membrane phospholipid levels.** Histology scores (**a**) and representative micrographs (**b**) in mice subjected to 60′ of ischemia +/− the Akt inhibitor GDC-0068 (50 mg/kg). $N = 6$ mice/group except for WT I60′ ($N = 5$); bars = 110 µm. Analyzes by Kruskal–Wallis with multiple correction and two-tailed Mann–Whitney $U$ test where appropriate. Non-GDC-0068 data reproduced from Fig. 2c. **c** Enteroid overnight hypoxia CTG assay of 7d enteroids ± 10 µM IWR-1-endo. Results pooled from two independent experiments, total $N = 20$/group. $p < 0.0001$ (−IWR), $p = 0.0487$ (+IWR) by one-way ANOVA with multiple comparison correction. **d** 100 µM IWP-2 (a Wnt inhibitor) or 10 µM IWR-1-endo (an Axin2-stabilizing, direct β-catenin inhibitor) was added simultaneously to growth factor withdrawal and spheroid generation were observed. No $Tipe0^{-/-}$ enteroids were observed with IWR-1-endo. $N = 2$ independent experiments for IWP-2, $N = 3$ for IWR-1-endo; images representative of 3-5 spheroids per independent experiment; bars = 70 µm, IF as shown. Survival shown as percent; $P = 0.0001$ for IWP-2 and $p < 0.0001$ for IWR-1-endo by Mantel–Cox test. **e** Extraction of BODIPY-TMR-tagged PIP2 (Ptdlns(4,5)P$_2$) or PIP3 (Ptdlns(3,4,5)P$_3$) in the presence of PIP3 or PIP2 (respectively) by TIPE0 from BODIPY-FL containing vesicles, measured by a decrease in FRET signal. All samples normalized to non-extracting PLCδ-PH protein. Negative extraction indicates binding without extraction (enhanced FRET efficiency vs control). $N = 9$ pooled across three independent experiments (PIP2 studies); $N = 6$ pooled across two independent experiments (PIP3 studies). Analyzes by two-tailed $t$-test. **f** Mass ELISA of whole gut tissues for PIP3, normalized to equivalent WT levels, ±antibiotics. Number of mice per group as follows: WT & $Tipe0^{-/-}$ colons = 8, WT & $Tipe0^{-/-}$ colons+abx = 4, WT and $Tipe0^{-/-}$ ileums = 6, WT ileum+abx = 3, $Tipe0^{-/-}$ ileum +abx = 4. Analyses by two-tailed Mann–Whitney $U$ test. **g, h** Co-localization analysis of Akt-PH-GFP (sensing Ptdlns(3,4,5)P$_3$) with the cell membrane in NC (negative control) and TKD ($Tipe0$-knockdown) CMT-93 cells (**h**). $N = 26$/group, pooled from 3 independent experiments. $p < 0.0001$ by two-tailed $t$-test. Representative images shown in **g**; bars = 10 µm. For all graphs, error bars show mean ± SEM; $*p < 0.05$; $**p < 0.01$; $***p < 0.001$; $****p < 0.0001$. Source data are provided as a source data file.

synthesized by random hexamer-primer using the fragments as templates, and dTTP was substituted by dUTP during the synthesis of the second strand. Short fragments were purified and resolved with elution buffer for end reparation and single nucleotide A (adenine) addition. After that, the short fragments were connected with adapters, then the second strand was degraded finally using UNG (Uracil-N-Glycosylase) [2]. After agarose gel electrophoresis, the suitable fragments were selected for the PCR amplification as templates. During the quality control steps, Agilent 2100 Bioanaylzer (https://www.genomics.agilent.com/en/Bioanalyzer-System/2100-Bioanalyzer-Instruments/?cid=AG-PT-106) and ABI StepOnePlus Real-Time PCR System (https://www.thermofisher.com/order/catalog/product/4376600) were used in quantification and qualification of the sample library. At last, the library was sequenced using Illumina HiSeq4000 using PE100 strategy. Primary sequencing data that produced by Illumina Hiseq4000 called as raw reads, were filtered into clean reads by remove adaptor contained and low-quality reads by BGI (Shenzhen, China) in-house software. Reference annotation-based assembly method was used to reconstruct the transcripts by Tophat (v2.0.10) + Cufflinks (v2.1.1), while background noise was reduced by using fragments per kilobase million (FPKM) and coverage threshold.

Data analysis was carried out using the statistical computing environment, R, the Bioconductor suite of packages for R, and RStudio (https://www.rstudio.com/)[40]. Raw data were background subtracted, variance stabilized, and normalized by robust spline normalization. Differentially expressed genes were identified by linear modeling and Bayesian statistics using the Limma package. Probes sets that were differentially regulated after controlling for multiple testing using the Benjamini-Hochberg method were used for hierarchical clustering and heatmap generation in R. Clusters of co-regulated genes were identified by Pearson correlation using the hclust function of the stats package in R. Pathway analysis was performed using GSEA (http://software.broadinstitute.org/gsea/index.jsp). Briefly, GSEA is a computational method that determines whether a priori defined set of genes shows statistically significant, concordant differences between two biological states. GSEA does not focus on only significantly/highly changed genes but examines all the genes that belongs to a certain biological process instead.

**Single-cell RNASeq.** Enterocytes were isolated as above and then subjected to enzymatic digestion with 10 U/mL DNAse I and 1 U/mL Dispase for 15 min at 37 °C.

Single-Cell RNASeq was done using the Chromium System (10x Genomics, Pleasonton, CA) and the Chromium Single Cell 3′ Reagent Kits v2 (10x Genomics), in accordance with the manufacturer's protocol. Following sequencing, initial data processing of samples was performed using Cellranger (v.2.1.0, 10x Genomics). Cellranger mkfastq was used to generate demultiplexed FASTQ files from the raw sequencing data. Next, cellranger count was used to align sequencing reads to the mouse reference (mm10), and generate single cell gene barcode matrices. Cellranger aggr was used to aggregate the matrices and normalize by mapped read depth to account for sequencing depth. Post processing and secondary analysis of the aggregated dataset was performed using the Seurat package (v.3.0) within R (v.3.5.1). Variable features across single cells were identified by mean expression and dispersion. Identified variable features were used to perform a PCA. The first 14 PCs were used to cluster cells with default parameters and the reduced dimensions were visualized in a tSNE plot. Specific cell types were identified using top differentially expressed cell markers from Haber, et al.[41]. For comparisons of differentially expressed genes, each cluster were identified by Seurat function "find_all_markers" using a wilcoxon rank sum test. A Bonferroni correction was used to account for multiple testing based on the total number of genes in the dataset.

To determine enriched pathways and trends in the scRNAseq datasets we used gene set enrichment analysis (GSEA) and IPA. Average log fold change between samples (TKO vs. WT) was used in the gene pathway enrichment analyses. Specific gene sets used in the GSEA analysis included FEVR_CTNNB1_TARGETS_DN (M2343), FEVR_CTNNB1_TARGETS_UP (M2342), and ST_WNT_BETA_CATENIN_PATHWAY (M17761). Average log fold differences between samples were overlayed on the PI3k/Akt signaling pathway in IPA. This data was used to identify gene expression trends between samples affecting the PI3k/Akt signaling pathway.

**Flow cytometry.** Enterocytes were isolated as above and then fixed in fresh, ice-cold 1% PFA/PBS for 10 min. Cells were then washed twice with 2% FBS in PBS, then digested in a 10 ml PBS solution containing 2% FBS and 0.25 mg/ml collagenase IV for 30 min at 37 under shaking (200 rpm). Digested cells washed twice with 2% FBS and 2 mM EDTA in PBS, passed through 70 µM filter to collect single cells. Cells were stained 1:100 dilutions at RT for 20 min before washing. Isotype controls were used at 1:100 dilutions to determine the background caused by nonspecific staining. Cells were analyzed on a LSRII (BD Biosciences, San Jose, CA) according to the gating strategy (Supplementary Fig. 4b–e). Data were analyzed with FlowJo v10.0.7 (BD). All the experiments were independently repeated two times in triplicate. The following antibodies were used: Anti-mouse Cd326 Ep-CAM APC/FIRE (118229, Biolegend) and Brilliant Violet 421™ anti-mouse Ly-6A/E (d7) (108127, Biolegend).

**Microscopy and image processing.** Brightfield microscopy was performed on an Echo Revolve (Echo, San Diego, CA). Confocal images were gathered with a Leica TC8 SP8 Multiphoton Confocal (Leica Microsystems, Buffalo Grove, IL) or a Zeiss 810 (Carl Zeiss, Oberkochen, Germany). Images were processed with FIJI (version 1.51w, National Institutes of Health, Bethesda, MD). All images in a given panel were captured with identical settings for white light intensity, exposure/gain and light/laser intensity and were processed identically. Co-localization analysis was performed in FIJI using the Coloc2 Module and the spatial analysis first described by Li et al.[42]. For immunohistochemistry staining, white balancing was applied equally to all images in a panel using Adobe Photoshop.

**FRET extraction assay.** Recombinant TIPE2, TNFAIP8, and PLCδ-PH were expressed from *Escherichia coli* BL21(DE3) cells (Agilent, Santa Clara, CA) and purified using Ni-NTA Agarose (Qiagen, Germantown, MD). 6His-SUMO tagged proteins were eluted with 250 mM Imidazole from beads, followed by cleavage with SUMO Protease 1. The SUMO fusion proteins and SUMO Protease after cleavage were removed by affinity chromatography on a second Ni-chelating resin. Final eluates with untagged native proteins were concentrated using Amicon Ultra centrifugal filters (Sigma Millipore), and dialyzed in HBS (25 mM HEPES, 150 mM NaCl, pH 7.4) buffer using Slide-A-Lyzer cassettes (Thermo Fisher). The purified proteins were at least 95% pure as judged from overloaded Coomassie Blue G-250 stained SDS gels. Protein concentrations were determined based on absorbance at 280 nm using calculated extinction coefficients. Vesicles containing DOPC as a backbone, with 10% PC-BODIPY FL (Thermo fisher), and either 10% PIP2 ((Ptdlns(4,5)P$_2$) or PIP3-BODIPY-TMR (Echelon Biosciences, Salt Lake City, UT) with or without the addition of 10% PIP3 or PIP2 (respectively, from cellsignals. net) were made in the same protein dialysis buffer used for the protein purification, using the freeze/that/sonication method to create 50–100 nm vesicles. Vesicles and proteins were mixed to a final concentration of 1 mM vesicle and 20 µM protein

and incubated and room temperature. Samples were then diluted 1:10,000 in buffer to quench the reaction and decrease total fluorescence signal.

FRET signals were measured in a 4 mL quartz cuvette filled to 3 ml with stirring on a Fluorolog 2 spectrofluorometer (Horiba, Piscataway, NJ) thermostatted to 25 °C, an excitation wavelength of 430 nm, emission wavelengths of 510 and 570 nm, and 5 nm slits. Wavelengths were selected by performing control vesicle experiments with each fluorophore individually and in combination, in the absence of proteins. Percent extraction was calculated by linear extrapolation between the signal obtained with PLCδ-PH binding (set to 0%), and the signal observed, adjusted by a factor of two to account for the fact that the minimum sign (set to 50% because proteins cannot access the inner leaflet of the vesicle).

**PIP3 mass ELISA.** PIP3 Mass ELISA (Echelon) was performed according to the manufacturer's protocol. Values were normalized to total protein levels measured in a small portion of the total lysate obtained prior to lipid purification.

**CMT-93 CRISPR-mediated TIPE0 knockdown.** $2 \times 10^6$ CMT-93 cells were transiently transfected with 2.5 μM Alt-R crRNA:tracrRNA$^{(ATTO\ 550)}$ complex and 200 nM Alt-R HiFi Cas9 Nuclease enzyme (IDT, Coralville, IA) using the Amaxa (Lonza, Basel, Switzerland) nucleofector II system and Kit R per the manufacturer's protocols. 24 h after transfection cells were sorted and positive cells (via GFP) were single-cell suspended in 96-well plates to form colonies. Following 4 weeks in culture, single cell colonies were trypsinized and divided into half. Half of the cells were sub-cultured in 48-well plates for further growth and the other half was subjected to DNA isolation. Genomic DNA was extracted using QuickExtract DNA Extraction Solution (Epicentre Biotechnologies, Madison, WI). Common PCR was conducted to amplify the target region using primers flanking the targets region (forward: 5′TGTGAGCCTGGCAACATAGG-3′, reverse: 5′-AAGGGATTGTAC AAGGCAGC-3′). The amplified regions were then subjected to Sanger sequencing. Percent of insertions and deletions (indels) in the DNA of the targeted region was further calculated using the TIDE webtool (https://tide.deskgen.com) relative to control cells. The crRNA used to target TIPE0 was pre-designed by IDT, labeled TNFAIP8.1.AA, targeting: 5′TGACGACCGTCATGGCAAGC-3′. The positive crRNA was IDT Alt-R CRISPR-Cas9 positive control Mouse HPRT, and the negative control crRNA was IDT Alt-R CRISPR-Cas9 negative control #1. TIPE0 expression was further evaluated by Western blot. The three clones with the lowest expression by Western and the highest indel rates were pooled to make the final polyclonal cell populations used for further studies. Three negative control samples with WT-levels of TIPE0 expression were similarly pooled.

**CMT-93 growth and death assays.** For the growth assays, parallel 96-well plates (NC and TKD) were seeded with equal numbers of cells for 1–5 days and a CTG assay was performed to determine growth over time, as described above. All samples were normalized to 1 day of growth. Death was measured by LDH (lactate dehydrogenase) levels in the collected culture media of the same cells analyzed in the CTG assay. LDH levels were measured using the CyQUANT LDH Cytotoxicity Assay (ThermoFisher Scientific) with a Tecan Infinite M200 PRO, per the manufacturer's protocol.

**Phospholipid membrane co-localization staining.** The PtdIns(3,4,5)P$_3$ in fixed cells was visualized using EGFP-tagged AKT-PH domain, which specifically binds to this phosphoinositide. PtdIns(4,5)P$_2$ was visualized using EGFP-tagged PLCδ-PH domain. Briefly, resting NC and TKD CMT-93 cells were transfected with GFP-C1-AKT-PH vector or GFP-C1-PLCdelta-PH vector (Addgene, Watertown, MA) using the mouse Nucleofector Kit R and an Amaxa Nucleofector II (program Y-001, Lonza). Cells were cultured for 24 h after the transfection and then fixed in 1% PFA for 10 min. Cells were washed in HBSS and then stained with Wheat Germ Agglutinin CF640R (Biotium, Fremont, CA). After several more washes cells were mounted in media containing DAPI and imaged on a Zeiss 810, before being subjected to colocalization analysis, as described above.

**Quantification and statistical analysis.** All two-group in vivo comparisons were performed by the Mann-Whitney U test. Two-group in vitro comparisons were by two-tailed student t-test. In vivo multiple group comparisons were by Kruskal–Wallis one-way ANOVA, with Dunn's multiple comparison post-test. Mann–Whitney U test was used to confirm ANOVA findings. Survival curves were by Mantel–Cox ranked test. Analyses were performed in Prism versions 7 & 8 (Graphpad, San Diego, CA). Statistical analyses for the RNA-Seq and scRNA-Seq are described above. All graphs are presented as mean ± SEM.

**Reporting summary.** Further information on research design is available in the Nature Research Reporting Summary linked to this article.

## Data availability
The authors declare that all data supporting the findings of this study are available within the article and its supplementary information files or from the corresponding author upon reasonable request. The sc-RNA-Seq datasets generated and/or analyzed during the current study have been deposited in GEO, under accession codes: GSE143916 [https://www.ncbi.nlm.nih.gov/geo/query/acc.cgi?acc=GSE143916], GSM4276625 [https://www.ncbi.nlm.nih.gov/geo/query/acc.cgi?acc=GSM4276625], GSM4276626 [https://www.ncbi.nlm.nih.gov/geo/query/acc.cgi?acc=GSM4276626]. The bulk RNA-Seq datasets generated and/or analyzed during this study have been deposited in the ArrayExpress database, under accession code E-MTAB-8796 [https://www.ebi.ac.uk/arrayexpress/experiments/E-MTAB-8796/]. DEGs from these data have been included as Supplementary Data 1 and 2. The source data underlying all figures has been provided as a Source Data File.

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

## Acknowledgements
We would like to thank University of Pennsylvania Perelman School of Medicine Cell and Developmental Biology Core, the Molecular Profiling Core, the Center for Molecular Studies in Digestive and Liver Diseases' Molecular Pathology and Imaging Core, and the Center for Applied Genomics NGS core at the Children's Hospital of Philadelphia for 10× compatible (scRNAseq) library preparation and sequencing. We would also like to thank Drs. Anil Rustgi and Sarah Andres for training in enteroid culture techniques, and Dr. Jerrold Turner for his Brdu/Edu pulse chase protocol. We would also like to thank Drs Paul Axelsen and Hiro Komatsu for their assistance with the FRET experiments, use of their fluorometer, and review of the manuscript.

## Author contributions
JRG designed, executed, and analyzed experiments and wrote the manuscript. NS executed and analyzed experiments and helped write the manuscript. NS, AZ, RH, ER, ML, TC, ZE, LL, MBD, JR Guzman, and AB designed, executed, and analyzed experiments. XL and MG analyzed RNA-Seq data. HS provided reagents and oversaw animal husbandry. AA performed the RNAScope studies under the guidance of JLW; both edited the manuscript. HH supervised the sc-RNA-Seq analysis. YHC supervised this study and helped write the manuscript.

## Competing interests
YHC is a member of the advisory boards of Amshenn Co. and Binde Co. None of the other authors have any competing interests to disclose.
