## [Peer Review File · Nature Communications]

Reviewers' Comments:

Reviewer #1:

Remarks to the Author:

Building off of earlier findings (Honghong et al, J Immunol, 2015), the authors in the current manuscript focus on the role of TNFAIP8 in acute injury and show that TNFAIP8-deficient epithelial cells are resistant to tissue damage and have regenerative deficits. Mechanistically, loss of TNFAIP8 results in increased microbiome-driven basal Akt activation and downstream β -catenin activation. Although the concept of this study is intriguing, there are some addressable weaknesses in the interpretation of some experiments (e.g., organoid cultures and immunostaining evaluation). Also, the discrepancy of decreased numbers of fetal-like cells and the increased expression of regenerative stem cell markers in Tipe0^{-/-} epithelial cells should also be further investigated. However, the authors' general contention that increased microbiome-driven basal Akt activation by loss of TNFAIP8 is probably an important contributor to the noted phenotype. Overall, the manuscript is well-written and the data support the conclusions. Nonetheless, several concerns need to be addressed in order for the manuscript to be suitable for publication.

Major points

1. In Fig 2, we recommend that the authors evaluate the survival of the organoids more carefully. For example, in Fig 2D, the authors only show images of one organoid in each group and there seems to be no difference between WT and Tipe0^{-/-} organoid images. Lower magnification images would be appreciated. Also, in Fig 2E, the authors claim that novel spheroids are established after media withdrawal; however, they do not show whether these spheroids arise from established previous organoids, which makes it hard to interpret these data. The discussion regarding this observation is a bit confusing.
2. On page 8, the authors mention the discrepancy of decreased number of fetal-like cells and increased expression of regenerative stem cell markers in Tipe0^{-/-} epithelial cells. From this finding, they conclude that a proper regeneration program is not engaged. The authors need to clarify this point or investigate this mechanism further.
3. In Fig 5, the authors state that loss of Tipe0 results in upregulated Akt signaling in a microbiota-dependent manner. These conclusions are largely supported by pAkt immunostains in Fig 5D and E. However, the background staining for Tipe0^{-/-} mice are much stronger compared to that of WT or antibiotic depletion mice (note background staining of muscularis propria in Tipe0^{-/-} intestines), which may preclude proper evaluation of any reported difference. Also, in Fig 5C, there is no band for β -catenin by immunoblotting of WT enterocytes, but there is a band for β -catenin by immunoblotting for WT enterocytes in Fig. 5F—clarification/discussion is needed for these differences. Do commensals also regulate β -catenin expression? Perhaps assessing β -catenin in crypts versus villi would be helpful as WT crypts should also have robust β -catenin staining. Is the β -catenin accumulation nuclear, cytoplasmic or both in vivo?
4. In Fig E5A, TIDE indel analysis shows insertions in the negative control gRNA clone, can the authors explain this should be explained.

Minor points

1. In Fig 3A, alkaline phosphatase staining cannot be recognized in the epithelial cells of both WT and Tipe0^{-/-} organoids.
2. In Fig 5A, immunostaining for β -catenin does not show nuclear staining. The authors should evaluate the number of cells with nucleus staining.

Reviewer #2:

Remarks to the Author:

Goldsmith and coauthors present a timely study examining the process of intestinal epithelial regeneration. Specifically, the authors focused on the role of TNF-alpha induced protein 8-like (TIPE), a lipid transport protein that regulates PI3K-mediated signaling. The authors examined the role of TIPEO in intestinal injury and show that loss of TIPEO results in resistance to epithelial injury. The authors show that TIPEO is a regulator of intestinal regeneration by functioning to inhibit activation of Akt, downstream B-catenin accumulation, and subsequent induction of B-catenin target genes. They demonstrate that TIPEO regulates Akt signaling by extracting PIP2 from the membrane, reducing the amount of PIP3 produced and thus inhibiting availability of PIP3 to activate the Akt signaling cascade. The authors suggest that this aberrant activation of Akt is responsible for inhibition of YAP-driven fetal-reprogramming in regeneration, although the link between Akt and YAP was not directly shown.

Overall, this is an interesting study focused on intestinal epithelial regeneration. Much of the proposed pathways seem to corroborate previously established links (e.g. ability of TIPE proteins to suppress PI3K signalling, etc...), with the exception of TIPEO removing PIP2 from the membrane which is a novel observation. Additionally, the major novelty of this study appears to lie in the findings that TIPE knockout mice are resistant to injury in association with a hyperproliferative response that results in a lack in epithelial differentiation and a deficiency of regenerative/reserve stem cells, suggesting that TIPEO is important in regulating the stem cell niche.

Although, the authors conducted a significant amount of experiments to support most of their conclusions, there are some significant shortcomings that make it difficult to accept the manuscript in its current form. Some major concerns include:

- 1) A major problem with the paper is the consistent lack of adequate image quality and resolution throughout. Although there is quantification provided, many of the histologic and fluorescent stained images are low power images with the stained cells being difficult to see and assess (examples of inadequate images include Fig 2L, 3C, 3F, entire Fig4, Fig 5A, etc). This makes interpretation of the results impossible and needs to be modified. Inclusion of higher power images, perhaps even including arrows to highlight what the reader should be seeing would be very helpful.
- 2) In the results section, the authors sometimes make definitive statements that seem a bit overreaching. For instance, on page 6 when referring to "TipeO^{-/-} enteroids surviving exposure to TNF α compared to WT", as it is currently written, the authors seem to imply that WT organoids don't survive at all while Tipe^{-/-} organoids do. In reviewing Fig 2A and 2B, however, both WT and TipeO^{-/-} organoids were susceptible to TNF α or radiation induced toxicity but TipeO^{-/-} organoids have a ~10% increase in survival, albeit this is a significant difference.
- 3) The authors make the claim that loss of TIPEO results in increased Akt in a microbiome driven response. This appears to be a major point in the discussion yet the data provided to make this conclusion appears weak. The data provided to support this claim, from what I can tell is simply Figure 5E and F in which antibiotics resulted in the lack of pAktS473 staining. This observation is weak at best. First, there is no positive control for the pAktS473 staining shown in the antibiotic experiment so it's hard to say if this was simply a technical limitation of the study or truly an absence of pAktS473 staining (this applies to both 5E and 5F). Second, to make the claim that "TIPEO normally inhibits microbiome-induced Akt activation" based on IHC staining and a Western for pAktS473 is not sufficient. If this is going to be a major point of the paper, then it would be nice to see other measures of Akt activity in antibiotic treated WT and TIPEO mice? Thirdly, is there a difference between the WT control in Fig 5B-C and 5E-F? Because it seems that B-catenin is absent in 5C but present and detectable in 5F?
- 4) Another important point is it would be nice to know what happens to TIPEO expression in WT mice during injury? If this is known, then the authors should at least discuss the data, and if not this would be nice to include as a rationale to demonstrate its importance in regeneration.

Minor points to address include:

5) Switching between analysis of mice/enteroids at baseline and during various types of injury (GF withdrawal, DSS, irradiation, I/R, etc...) impeded the logical flow of the paper. Could data perhaps be rearranged to make the comparison between homeostasis and injury more smooth. Additionally, clarification of what growth factor removal does would be helpful. This was not clear from the results or Figure Legends.

6) In Figure 1:

o Authors claim there is no change in cytokine levels but some of the cytokines look very different between knockout groups

o The claim that protection in Tipe0^{-/-} mice is strictly epithelial-dependent is weak as it is ubiquitously expressed and although a BMTx experiment was alluded to, this does not prove that it is strictly epithelial and perhaps this statement can be softened (i.e. suggests that...)

7) In Figure 2:

o In Figure 2D, the organoids look the same in WT and KO organoids across all time-points (i.e. both groups look like they are dying)

♣ How was survival quantified? Methods section suggests that they used CellTitreGlo – data for this?

♣ Better quality images would significantly improve figure.

o 2J) Unclear what is being shown with white arrows

o Unclear why some Tipe0^{-/-} organoids are unable to grow but others are hyperproliferative – seems paradoxical and not really addressed

8) In Figure 3:

o 3F) Would have liked to see quantification of data for #Paneth cells/crypt-villus unit.

♣ 3G) Hard to interpret data here but would be nice if authors could better describe what they are trying to show – i.e. in particular where are the Lgr5⁺ cells and Fetal-like cells because as it stands, I cannot see either population?

o 3G-H) They claim decreased levels of differentiated lineages in Tipe0^{-/-} mice – however there seems to be an increase in EE cells and goblet cells? Does this not go against the claim there are less differentiated cells with Tipe0^{-/-} mice?

9) In Figure 4:

o The authors claim that Tipe0^{-/-} mice have inappropriate basal activation of the Sca1⁺ injury response but there is no link to Yap shown. For instance, it would be nice to know if basal Yap is upregulated in these mice as well?

o Unclear explanation of the discrepancy between fewer Sca1⁺ cells in Tipe0^{-/-} mice and what seems to be activation of the program at baseline

10) In Figure 5:

o 5A) Overall KO image looks brighter here, not just the green – altered exposure of images confounding results?

o Good link showing loss of Tipe0 upregulates pAkt and downstream B-catenin which are both reduced with antibiotic treatment

o

11) Extended figures:

o Data with missing quantification in many of these figures

12) The authors provided data showing that the Wnt antagonist IWP-2 did not block the formation of Sca1⁺ spheroids but IWR-1-endo (promotes B-catenin degradation) abolished all Sca1⁺ spheroids that appeared upon GF withdrawal, indicating that non-Wnt-dependent B-catenin accumulation was responsible for development of the Sca1⁺ spheroids. In this setting, do the authors know what is then responsible for B-catenin accumulation if not Wnt?

Reviewer #3:
Remarks to the Author:
Remarks to the Author:

In the current manuscript the authors reported a potential role of TNFAIP8/Tipe0 in inactivating PI3K/Akt signaling in IECs. In addition, the authors suggested that in the absence of TNFAIP8/Tipe0 the enterocytes are more resistant to injury with a hyperproliferative phenotype. Furthermore, the authors observed strong defects in the differentiation process in the intestinal epithelium (strong induction of Clu+ regenerative program and a marked increase in the expression of Sca1) when TNFAIP8/Tipe0 was missing. Furthermore, upon injury TNFAIP8/Tipe0 deficiency resulted in a defective epithelial regeneration.

While the topic in principle is interesting unfortunately at the current stage the manuscript lacks significantly in quality and structure and appears rather premature. In general, the experimental approach is correct and well described but the rationale and the results from those experiments do not support the conclusions drawn here. Therefore the data presented here need further strengthening to support conclusions made.

CRITIQUES.

1. Figure 1. Injury score for healthy KO animals must be shown (Figure 1A). In figure 1B, representative images of the intestinal mucosa in healthy KO animals are missing. Incorporating low magnification images in figure 1B must be considered. Cytokine expression, secretion and production must be evaluated in the intestinal mucosa of healthy KO animals and not only after I/R. The expression and localization of Tipe0 and Tipe2 proteins needs to be evaluated in the whole mucosa and in freshly isolated crypts of healthy and I/R WT mice (IHC and RT-PCR, could be used). In figure E1, proper controls using healthy KO animals are also missing. Quantitation in figure E1D is necessary and the analysis for Tipe0 should be incorporated. In figure E1D, the use of specific markers to evaluate different immune cell populations present in the intestinal mucosa is highly recommendable. The images used in figure E2 are poor quality (Tunel and active caspase 3 staining) should be replaced. Quantitation is also required and must be performed in healthy conditions for both, Wt and KO animals. Why is some Tunel staining observed in healthy adjacent tissue in figure E2C? Why the image presented as WT I/R90 in figure E2B is so different to the figure used as WT I/R90 in panel 1B, low magnification images with high magnification insets should be more appropriate to illustrate both panels.

2. Important: Authors should consider eliminating experiments carried out with Tipe2 animals.

3. Figure 2. Results in figure 2 are very interesting but difficult to connect. E.g. in the absence of Tipe0 IECs are resistant to injury, more proliferative but displayed less intestinal epithelial stem cells per unit area. The images presented in Figure 2J are very poor quality. Authors need to clarify if the treatment with TNF was carried out overnight (Text) or for 24h (Figure and figure legend). The incorporation rate for EdU and BrdU at basal states in WT and Tipe0 KO animals should be evaluated and quantitation presented. In figure 2L, the pulse and chase experiments suggest a high rate turnover of IECs lacking Tipe0, is apoptosis increased in those animals? Is the crypt length different in WT and Tipe0 KO animals? Those parameters must be evaluated and the results incorporated in the manuscript.

4. Figure 3 and 4. Is difficult to understand the logical behind the experiments presented in both figures. E.g. why the authors analyzed the expression/presence of alkaline phosphatase and keratin 20 in Wt and Tipe0-/- organoids? Why the authors used DSS induced colitis as a model in figure 4, given that DSS mainly affects the colon and the experiments in Figure 1 and 2 are in small intestine. Also, a clear connection between the findings reported in figure 3 and 4 and the results reported in figure 1 and 2 is missing. In figure 4, the use of Lgr5 reporter mice instead of Lgr5 antibodies is highly

recommended. IF images in figure 4 must be improved. Figure 3A needs to be improved/replaced.

5. Figure 5 and 6. The results in figure 1 and 2 are clearly linked to the changes induced in Akt signaling (figures 5 and 6) by the absence of Tipe0 protein, however the connection established with β -catenin signaling is confusing. TNFAIP8/Tipe0 absence drives Akt hyperactivation but did not affect Erk 1/2 signaling (Figure 5 and data not shown), the effect of Tipe0 in Erk must be shown.

Response to Reviewers

We thank all reviewers for their insightful analyses and constructive comments. We are pleased to learn that all reviewers were highly enthusiastic about the significance and innovation of this work, and their concerns related primarily to the scope and the discussion of the results. In this revised manuscript, we have addressed their concerns thoroughly, either by performing the suggested experiments or by supplying the requested information. We believe that this revision further enhanced the impact of this work and helped establish conclusively that TIPE0 modulates intestinal homeostasis by regulating the steady-state levels of PIP3 generated by commensal microbiota-induced PI3K activity and associated downstream signaling (namely Akt activation). Summarized below are our point-by-point responses to the reviewers' comments (shown in **blue font**). Major changes are in **red** in the main manuscript.

Reviewer #1 (Remarks to the Author):

Building off of earlier findings (Honghong et al, J Immunol, 2015), the authors in the current manuscript focus on the role of TNFAIP8 in acute injury and show that TNFAIP8-deficient epithelial cells are resistant to tissue damage and have regenerative deficits. Mechanistically, loss of TNFAIP8 results in increased microbiome-driven basal Akt activation and downstream β -catenin activation. Although the concept of this study is intriguing, there are some addressable weaknesses in the interpretation of some experiments (e.g., organoid cultures and immunostaining evaluation). Also, the discrepancy of decreased numbers of fetal-like cells and the increased expression of regenerative stem cell markers in Tipe0^{-/-} epithelial cells should also be further investigated. However, the authors general contention that increased microbiome-driven basal Akt activation by loss of TNFAIP8 is probably an important contributor to the noted phenotype. Overall, the manuscript is well-written and the data support the conclusions. Nonetheless, several concerns need to be addressed in order for the manuscript to be suitable for publication.

Major points

1. In Fig 2, we recommend that the authors evaluate the survival of the organoids more carefully. For example, in Fig 2D, the authors only show images of one organoid in each group and there seems to be no difference between WT and Tipe0^{-/-} organoid images. Lower magnification images would be appreciated. Also, in Fig 2E, the authors claim that novel spheroids are established after media withdrawal; however, they do not show whether these spheroids arise from established previous organoids, which makes it hard to interpret these data. The discussion regarding this observation is a bit confusing.

Response: *We have added our CellTiterGlo data to our survival curve to strengthen the analysis of the survival data presented in Figure 2D. We have changed the images of the time course in Fig 2D to better demonstrate our findings. Regarding the spheroids that develop, these appear de novo in the same ongoing culture one day after withdrawal of all growth factor supplementation, usually from very early budding crypts. This is clarified in the manuscript (Page 6), and we have replaced the images in Fig 2E to better demonstrate this.*

2. On page 8, the authors mention the discrepancy of decreased number of fetal-like cells and increased

expression of regenerative stem cell markers in Tipe0^{-/-} epithelial cells. From this finding, they conclude that a proper regeneration program is not engaged. The authors need to clarify this point or investigate this mechanism further.

Response: We have clarified this point further in the text (Page 8); specifically, we clarified that we don't see an increase in "4+ stem cell markers" which is different from the fetal-like cells and their markers. We also further explain that the discrepancy between a subset of the markers and overall number of cells is part of the dysregulation we see—the program is partially turned on erroneously but not fully functional.

3. In Fig 5, the author state that loss of Tipe0 results in upregulated Akt signaling in a microbiota-dependent manner. These conclusions are largely supported by pAkt immunostains in Fig 5D and E. However, the background staining for Tipe0^{-/-} mice are much stronger compared to that of WT or antibiotic depletion mice (note background staining of muscularis propria in Tipe0^{-/-} intestines), which may preclude proper evaluation of any reported difference. Also, in Fig 5C, there is no band for β -catenin by immunoblotting of WT enterocytes, but there is band for β -catenin by immunoblotting for WT enterocytes in Fig. 5F—clarification/discussion is needed for these differences. Do commensals also regulate β -catenin expression? Perhaps assessing β -catenin in crypts versus villi would be helpful as WT crypts should also have robust β -catenin staining. Is the β -catenin accumulation nuclear, cytoplasmic or both in vivo?

Response: We completely agree that the muscularis propria pAktS473 staining for the healthy Tipe0^{-/-} mice is much stronger compared to that of the WT and antibiotic depletion mice; however, this is a real effect, and not differences in background staining. We even seen it in our irradiated mice, but to a lesser extent. All slides were stained for the same amount of time, and furthermore we see this increased muscularis staining as compared to IgG control. We have included the IgG control as part of Figure S1, low magnification images of our irradiated mice to Figure E4, and added additional text commenting on this global elevation in signal. Additionally, we have repeated the pAkt IHC staining with positively staining controls and now compare them side by side in Figure 5E.

The differences in WT control for Fig5C and Fig5F were due to experiment-to-experiment variations in the Western blotting procedure (e.g. total protein loaded, transfer efficiency, etc.). A different Western with more equivalent baselines has been used in the resubmission.

We quantified the strength of staining in both the crypts and villi of healthy mice, mice subjected to ischemic injury, and after antibiotic administration, and found that the Tipe0^{-/-} mice had increased staining in the villi and more of this staining was nuclear, while antibiotics ablated these differences. This data is presented in Fig E5C&H. We also performed nuclear colocalization analysis for the enteroid β -catenin staining (see Minor Point 2, below), and saw increased nuclear staining with loss of TIPE0 in that experiment. These changes β -catenin levels and nuclear localization with loss of Tipe0 correlate well with changes in the transcription of β -catenin-dependent genes (Fig E5d-f) we have reported.

The addition of the β -catenin staining in the presence and absence of antibiotics was added to demonstrate that we can see downstream consequences of alterations in Akt activation in the presence or absence of antibiotics in WT vs Tipe0^{-/-} genetic backgrounds.

4. In Fig E5A, TIDE indel analysis shows insertions in the negative control gRNA clone, can the authors explain this should be explained.

Response: *We appreciate the reviewers catching this error. The wrong image was mistakenly inserted, which had shown a positive, non-specific control. We have replaced this with the proper image (now in Fig E6A).*

Minor points

1. In Fig 3A, alkaline phosphatase staining cannot be recognized in the epithelial cells of both WT and Tipe0^{-/-} organoids.

Response: *We have removed Figure 3a, as it was causing significant confusion and not adding a lot of value.*

2. In Fig 5A, immunostaining for β -catenin does not show nuclear staining. The authors should evaluate the number of cells with nucleus staining.

Response: *We agree it is hard to detect the nuclear fragment of the β -catenin staining given the strength of the DAPI channel. We performed colocalization analysis to directly address if nuclear staining was present, we did indeed see more nuclear staining with loss of TIPEO. This has been added to Fig 5C.*

Reviewer #2 (Remarks to the Author):

Goldsmith and coauthors present a timely study examining the process of intestinal epithelial regeneration. Specifically, the authors focused on the role of TNF-alpha induced protein 8-like (TIPE), a lipid transport protein that regulates PI3K-mediated signaling. The authors examined the role of TIPEO in intestinal injury and show that loss of TIPEO results in resistance to epithelial injury. The authors show that TIPEO is a regulator of intestinal regeneration by functioning to inhibit activation of Akt, downstream B-catenin accumulation, and subsequent induction of B-catenin target genes. They demonstrate that TIPEO regulates Akt signaling by extracting PIP2 from the membrane, reducing the amount of PIP3 produced and thus inhibiting availability of PIP3 to activate the Akt signaling cascade. The authors suggest that this aberrant activation of Akt is responsible for inhibition of YAP-driven fetal-reprogramming in regeneration, although the link between Akt and YAP was not directly shown.

Overall, this is an interesting study focused on intestinal epithelial regeneration. Much of the proposed pathways seem to corroborate previously established links (e.g. ability of TIPE proteins to suppress PI3K signalling, etc...), with the exception of TIPEO removing PIP2 from the membrane which is a novel observation. Additionally, the major novelty of this study appears to lie in the findings that TIPE knockout mice are resistant to injury in association with a hyperproliferative response that results in a lack in epithelial differentiation and a deficiency of regenerative/reserve stem cells, suggesting that TIPEO is important in regulating the stem cell niche.

Although, the authors conducted a significant amount of experiments to support most of their conclusions, there are some significant shortcomings that make it difficult to accept the manuscript in its current form. Some major concerns include:

1) A major problem with the paper is the consistent lack of adequate image quality and resolution throughout. Although there is quantification provided, many of the histologic and fluorescent stained images are low power images with the stained cells being difficult to see and assess (examples of inadequate images include Fig 2L, 3C, 3F, entire Fig4, Fig 5A, etc). This makes interpretation of the results impossible and needs to be modified. Inclusion of higher power images, perhaps even including arrows to highlight what the reader should be seeing would be very helpful.

Response: We thank the reviewer for this important feedback. We have made sure all our images in the resubmission include both low and higher power images and have added arrows or other markings where we felt it would be helpful. We have also eliminated the file compression issues that occurred with the initial submission. In particular, Figure 4 has new inserts, with arrows as needed. We have moved the enteroid YAP, RNAScope images, and related analyses to Figure E4 to make room for these larger images and inserts. We have enlarged the images in Figure 5A, added high magnification inserts, and used dashed lines and arrows to emphasize staining for pAktS473 and β -catenin, respectively. Figures 2L, 3C, and 3F have also been modified.

2) In the results section, the authors sometimes make definitive statements that seem a bit overreaching. For instance, on page 6 when referring to “TipeO^{-/-} enteroids surviving exposure to TNF α compared to WT”, as it is currently written, the authors seem to imply that WT organoids don’t survive at all while Tipe^{-/-} organoids do. In reviewing Fig 2A and 2B, however, both WT and TipeO^{-/-} organoids were susceptible to TNF α or radiation induced toxicity but TipeO^{-/-} organoids have a ~10% increase in survival, albeit this is a significant difference.

Response: We see a roughly 2-fold improvement in survival of the TipeO^{-/-} vs the WT enteroids exposed to TNF (90% vs 56%) and 50% greater survival after hypoxia (37% vs 23%). We have clarified this in the text and edited the figures to make this easier to see and made sure our language was appropriate for the magnitude of change seen (Page 6).

3) The authors make the claim that loss of TIPEO results in increased Akt in a microbiome driven response. This appears to be a major point in the discussion yet the data provided to make this conclusion appears weak. The data provided to support this claim, from what I can tell is simply Figure 5E and F in which antibiotics resulted in the lack of pAktS473 staining. This observation is weak at best. First, there is no positive control for the pAktS473 staining shown in the antibiotic experiment so its hard to say if this was simply a technical limitation of the study or truly an absence of pAktS473 staining (this applies to both 5E and 5F). Second, to make the claim that “TIPEO normally inhibits microbiome-induced Akt activation” based on IHC staining and a Western for pAktS473 is not sufficient. If this is going to be a major point of the paper, then it would be nice to see other measures of Akt activity in antibiotic treated WT and TIPEO mice? Thirdly, is there a difference between the WT control in Fig 5B-C and 5E-F? Because it seems that B-catenin is absent in 5C but present and detectable in 5F?

Response: We have attempted to address this point on several levels, up and down the signaling axis from Akt. At the level of Akt, we have repeated our staining with a positive control, and this is now shown in Figure 5e. We have also added downstream β -catenin staining and nuclear staining

quantification +/- antibiotics as Figure E5h, as well as downstream pYAPS127/YAP ratios and nuclear YAP accumulation in the presence of antibiotics (Figure E6a-c), to demonstrate the loss of increased Akt activation with antibiotics administration also effected the downstream pathways of interest in these studies (Page 11).

Upstream, we looked further at the role of TIPEO in modulating PI3K activity by its regulation of phospholipid levels. Using mass ELISA, we were able to demonstrate increased levels of ileal and colonic PIP3 in *Tipe0^{-/-}* mice that is lost with antibiotic ablation of the microbiome (Fig6F). Together, this demonstrates that TIPEO regulates microbiota-mediated basal PI3K activity and downstream signaling by sequestering the PIP2 substrate away from PI3K (Page 13).

Lastly, as discussed in the replies to reviewer 1, the differences in WT control for Fig5C and Fig5F were due to experiment-to-experiment variations in the Western blotting procedure (e.g. total protein loaded, transfer efficiency, etc.). A different Western with more equivalent baselines has been used.

4) Another important point is it would be nice to know what happens to TIPEO expression in WT mice during injury? If this is known, then the authors should at least discuss the data, and if not this would be nice to include as a rationale to demonstrate its importance in regeneration.

Response: This is an important question, and is something we have looked at but failed to include in the initial manuscript. It is now presented in Figure E4a. TIPEO is not acutely induced after ischemic injury (at either the 160' or 1/R90' timepoints) but is induced during the regenerative phase of radiation injury and after DSS-mediated injury, suggesting that its important during regeneration (Page 10).

Minor points to address include:

5) Switching between analysis of mice/enteroids at baseline and during various types of injury (GF withdrawal, DSS, irradiation, I/R, etc...) impeded the logical flow of the paper. Could data perhaps be rearranged to make the comparison between homeostasis and injury more smooth. Additionally, clarification of what growth factor removal does would be helpful. This was not clear from the results or Figure Legends.

Response: We thank the reviewer for this feedback. Prior to the initial submission of the manuscript, we tried several arrangements of the manuscript, including separating the homeostasis and injury components of the study, which was how we initially drafted the manuscript. Unfortunately, much of the homeostasis studies (specifically what is presented in Figure 3 and associated Extended Figures) is used to help explain the changes in injury response and regeneration and separating them out would make the logical flow even more difficult to follow.

We have clarified what growth factors were removed in both the results and methods (Page 6).

6) In Figure 1:

o Authors claim there is no change in cytokine levels but some of the cytokines look very different between knockout groups

o The claim that protection in Tipe0^{-/-} mice is strictly epithelial-dependent is weak as it is ubiquitously expressed and although a BMTx experiment was alluded to, this does not prove that it is strictly epithelial and perhaps this statement can be softened (i.e. suggests that...)

Response: We performed cytokine mRNA expression analysis on additional ischemic mice to increase our N and added them to the existing data, and further added healthy controls well. We saw significant inductions in inflammatory cytokines after ischemia, but between genotypes no significance was appreciated (Page 4). As reported previously (please see the corresponding results section in the manuscript for references), healthy Tipe2^{-/-} mice had increased expression levels of most cytokines as compared to WT mice, and healthy Tipe0^{-/-} had increases in basal Tnf. Comparing before and after ischemia, the level of Il1b and Il6 were elevated for Tipe0^{-/-} mice.

We have also softened the conclusions from the BMT statements, and instead state that it demonstrates that TIPE0 is likely acting via non-immune cells (Page 5).

7) In Figure 2:

o In Figure 2D, the organoids look the same in WT and KO organoids across all time-points (i.e. both groups look like they are dying)

♣ How was survival quantified? Methods section suggests that they used CellTiterGlo – data for this?

♣ Better quality images were significantly improve figure.

o 2J) Unclear what is being shown with white arrows

o Unclear why some Tipe0^{-/-} organoids are unable to grow but others are hyperproliferative – seems paradoxical and not really addressed

Response: We have improved the quality of all images and made sure file compression degradation has not occurred with the revised manuscript.

Regarding the survival studies in Figure 2D, we performed both CellTiterGlo (CTG) assays and manually counted living enteroids across time by eye, with enteroids maintaining 50% or more of their structure being considered alive. The CTG assay measures ATP levels, and over time the dying enteroids released ATP into the media, confounding the longitudinal CTG results after 1 day, which is why manual counting was used for longitudinal studies. We have included the 24-hour CTG data and clarified this in the text (Page 6). We have also changed the images of the time course in Fig 2D to better demonstrate our findings.

For Figure 2J, we removed the arrows, as they were not aiding the image.

The discrepancy we see in enteroid growth verses proliferation is one of initiation verses propagation (to borrow terminology from oncology). The Tipe0^{-/-} enteroids have deficiencies in initiating regeneration, which we first demonstrate in Figure 3 and then explore further in Figure E4, as YAP activation is needed for crypts to begin to regenerate and we show this is defective in our knockout. This is contrast to the hyperproliferation and increased cell turnover, which appears in only those enteroids that are able to initiate regeneration, as more of the cells appear to stay proliferative for longer (Figure 2J&K) and their turnover in the gut is faster (Figure 2L-N). We have clarified this point throughout the text (Page 10).

8) In Figure 3:

o 3F) Would have liked to see quantification of data for #Paneth cells/crypt-villus unit.

♣ 3G) Hard to interpret data here but would be nice if authors could better describe what they are trying to show – i.e. in particular where are the Lgr5+ cells and Fetal-like cells because as it stands, I cannot see either population?

o 3G-H) They claim decreased levels of differentiated lineages in Tipe0^{-/-} mice – however there seems to be an increase in EE cells and goblet cells? Does this not go against the claim there are less differentiated cells with TipeO^{-/-} mice?

Response: *We have added quantification of the Paneth cells. We have also added an insert demonstrating where the specific Lgr5+ and fetal-like cells are on the tSNE plot (they were not originally made visible because they are rare populations overlapping other regions), and have circled those regions on the main graph. We have clarified the text to read that there are less differentiated enterocytes in favor of transitional epithelial cells and secretory cells (EE and goblet cells), showing overall altered differentiation (Page 8).*

9) In Figure 4:

o The authors claim that Tipe0^{-/-} mice have inappropriate basal activation of the Sca1+ injury response but there is no link to Yap shown. For instance, it would be nice to know if basal Yap is upregulated in these mice as well?

o Unclear explanation of the discretion between fewer Sca1+ cells in Tipe0^{-/-} mice and what seems to be activation of the program at baseline

Response: *We have stained for baseline YAP staining, analyzed for nuclear colocalization (indicating active YAP), and done Western blots for total YAP and pYAPS127, the inactivated form of YAP. Together, this data (in Figure E4H-J) demonstrates less active YAP at baseline in Tipe0^{-/-} mice. YAP is phosphorylated by activated Akt at S127, also creating a direct link between the Akt and YAP signaling we see. Thus, the increase in Sca-1 is not YAP-mediated, which aligns with the post-injury data, as YAP activation drives the post-injury plastic response, and this response is lost with TIPE0-knockout. Sca-1 is also a β -catenin dependent gene, and yet high levels of β -catenin lead to perdurance of Lgr5⁺ cells and diminished YAP-mediated regeneration. Thus, the high levels of Akt/ β -catenin could be driving this inappropriate activation at baseline and simultaneously blocking Sca-1 induction and regeneration after injury. We have expanded upon this point in the discussion (Page 15, please see the main text for references).*

Thus, the fetal-like program appears partially activated at baseline (in particular Sca-1 levels are elevated), by single cell-RNASeq there are fewer true fetal-like cells at baseline, and the program does not induce after injury. This unique dysregulation—partial activation with fewer true fetal-like cells and a lack of a plastic response and induction after injury—is the key regenerative phenotype of the Tipe0^{-/-} mice, which appears to be due to the dysregulated signaling we describe. In follow-up work, we plan to explore how the high basal levels of Akt activation induce this dysregulated state in more detail. We have clarified this explanation in the manuscript (Page 8, 15).

10) In Figure 5:

o 5A) Overall KO image looks brighter here, not just the green – altered exposure of images confounding results?

Response: *The exposure was the same for all paired images in Figure 5. This is clearer with the higher magnification inserts we have included in the revised document.*

o Good link showing loss of Tipe0 upregulates pAkt and downstream B-catenin which are both reduced with antibiotic treatment

Response: *We greatly appreciate this positive feedback, thank you.*

11) Extended figures:

o Data with missing quantification in many of these figures

Response: *We have quantified all of the imaging data present in all of the extended Figures.*

12) The authors provided data showing that the Wnt antagonist IWP-2 did not block the formation of Sca1+ spheroids but IWR-1-endo (promotes B-catenin degradation) abolished all Sca1+ spheroids that appeared upon GF withdrawal, indicating that non-Wnt-dependent B-catenin accumulation was responsible for development of the Sca1+ spheroids. In this setting, do the authors know what is then responsible for B-catenin accumulation if not Wnt?

Response: *We believe activated Akt, through inactivation of GSK3 β , is leading to the β -catenin accumulation. We have clarified this point in the manuscript (Page 14).*

--

Reviewer #3 (Remarks to the Author):

Remarks to the Author:

In the current manuscript the authors reported a potential role of TNFAIP8/Tipe0 in inactivating PI3K/Akt signaling in IECs. In addition, the authors suggested that in the absence of TNFAIP8/Tipe0 the enterocytes are more resistant to injury with a hyperproliferative phenotype. Furthermore, the authors observed strong defects in the differentiation process in the intestinal epithelium (strong induction of Clu+ regenerative program and a marked increase in the expression of Sca1) when TNFAIP8/Tipe0 was missing. Furthermore, upon injury TNFAIP8/Tipe0 deficiency resulted in a defective epithelial regeneration.

While the topic in principle is interesting unfortunately at the current stage the manuscript lacks significantly in quality and structure and appears rather premature. In general, the experimental approach is correct and well described but the rationale and the results from those experiments do not

support the conclusions drawn here. Therefore the data presented here need further strengthening to support conclusions made.

CRITIQUES.

1. Figure 1. Injury score for healthy KO animals must be shown (Figure 1A). In figure 1B, representative images of the intestinal mucosa in healthy KO animals are missing. Incorporating low magnification images in figure 1B must be considered. Cytokine expression, secretion and production must be evaluated in the intestinal mucosa of healthy KO animals and not only after I/R. The expression and localization of Tipe0 and Tipe2 proteins needs to be evaluated in the whole mucosa and in freshly isolated crypts of healthy and I/R WT mice (IHC and RT-PCR, could be used). In figure E1, proper controls using healthy KO animals are also missing. Quantitation in figure E1D is necessary and the analysis for Tipe0 should be incorporated. In figure E1D, the use of specific markers to evaluate different immune cell populations present in the intestinal mucosa is highly recommendable. The images used in figure E2 are poor quality (Tunel and active caspase 3 staining) should be replaced.

Quantitation is also required and must be performed in healthy conditions for both, Wt and KO animals. Why is some Tunel staining observed in healthy adjacent tissue in figure E2C? Why the image presented as WT I/R90 in figure E2B is so different to the figure used as WT I/R90 in panel 1B, low magnification images with high magnification insets should be more appropriate to illustrate both panels.

Response: Images from healthy KO animals have been added, and low and high magnification histology has been included for both the healthy and ischemic tissue. We have also included healthy control tissue images and histological scoring for the chimeric mice. The healthy controls for the 160' experiments are the same for I/R90' experiments, and so these data were not presented again.

We performed cytokine mRNA expression analysis on additional ischemic mice and added them to the existing data, and further added healthy controls well. We saw significant inductions in inflammatory cytokines after ischemia, but between genotypes no significance was appreciated. As reported previously (please see the corresponding results section in the manuscript for references), healthy Tipe2^{-/-} mice had increased expression levels of most cytokines as compared to WT mice, and healthy Tipe0^{-/-} had increases in basal Tnf. Comparing before and after ischemia, the level of Il1b and Il6 were elevated for Tipe0^{-/-} mice.

Regarding the expression of TIPE0 and TIPE2, this has been previously reported and is available on The Human Protein Atlas (www.proteinatlas.org) and the EMBL-EBI gene expression atlas (www.ebi.ac.uk/gxa/home) and has been previously explored by us and others (see Goldsmith, et al. 2017). Specifically, TIPE2 is highly restricted to immune cells while TIPE0 is ubiquitously expressed. We have emphasized the existing knowledge in the text (Page 4). We further performed PCR and healthy and injured tissues (after ischemia, radiation, and DSS) to demonstrate expression changes in TIPE0 after injury, and this is included now as Figure E4A. TIPE0 is not acutely induced after ischemic injury (at either the 160' or I/R90' timepoints) but is induced during the regenerative phase of radiation injury and after DSS-mediated injury, suggesting that its important during regeneration. Unfortunately, the available TIPE0 and TIPE2 antibodies do not work with tissue IHC, and so we could not perform this analysis.

We have quantified the data for Figure E1D (now Figures E1E-F). We did not perform these studies in the Tipe0^{-/-} mice as our chimeric data and the Tipe2^{-/-} controls show that the effects of Tipe0^{-/-} are predominately non-immune, and the focus of this paper is on the non-immune effects of Tipe0. The

experiments of Figure E1D are to show that the known phenotype of altered immune cell migration in *Tipe2*^{-/-} mice is present here. While we agree further studies to identify the specific immune cell population altered by *Tipe2*-loss is an interesting question, that is the focus of other work by our group that has been published or currently under consideration for publication, and is beyond the scope of this manuscript.

The images in Figure E2 have all been replaced with higher quality images, as well as high magnification inserts. The TUNEL staining observed in the healthy tissue is from luminal contents, which are TUNEL positive. Regarding the differences in WT I/R90 between Figure 1B and Figure E2B, Figure 1B is a representative micrograph with a histological score near the average for that group. For Figure E2B, we chose a slightly less damaged tissue (score of 2.5 vs 3 for Figure 1B) in order to better demonstrate the staining differences. Highly damaged tissues are often so damaged that staining is difficult to appreciate. We have also quantified all the staining.

2. Important: Authors should consider eliminating experiments carried out with *Tipe2* animals.

Response: We thank the reviewer for this suggestion and strongly considered it. However, after reviewing the manuscript without the *Tipe2* data present, we feel that the addition of *Tipe2* serves as an important control. *Tipe0* and *Tipe2* have functional overlap in immune cells, and thus differences between the *Tipe0* and *Tipe2* knockouts help us elucidate the non-immune effects of loss of *Tipe0*. We thus decided to keep the data in this paper.

3. Figure 2. Results in figure 2 are very interesting but difficult to connect. E.g. in the absence of *Tipe0* IECs are resistant to injury, more proliferative but displayed less intestinal epithelial stem cells per unit area. The images presented in Figure 2J are very poor quality. Authors need to clarify if the treatment with TNF was carried out overnight (Text) or for 24h (Figure and figure legend). The incorporation rate for EdU and BrdU at basal states in WT and *Tipe0* KO animals should be evaluated and quantitation presented. In figure 2L, the pulse and chase experiments suggest a high rate turnover of IECs lacking *Tipe0*, is apoptosis increased in those animals? Is the crypt length different in WT and *Tipe0* KO animals? Those parameters must be evaluated and the results incorporated in the manuscript.

Response: The discrepancy we see in enteroid growth versus proliferation is one of initiation versus propagation (to borrow terminology from oncology). The *Tipe0*^{-/-} enteroids have deficiencies in initiating regeneration, which we first demonstrate in Figure 3 and then explore further in Figure E4, as YAP activation is needed for crypts to begin to regenerate and we show this is defective in our knockout. This is contrast to the hyperproliferation and increased cell turnover, which appears in only those enteroids that are able to initiate regeneration, as more of the cells appear to stay proliferative for longer (Figure 2J&K) and their turnover in the gut is faster (Figure 2L-N). We have clarified this point throughout the text.

We have clarified in the text that the TNF experiments were carried out for 24 hours. Higher quality images were used for Figure 2J, along with higher magnification inserts. Image quality was unfortunately degraded due to the compression needed for initial submission file size limits.

We have added Edu and Brdu incorporation rates as Figure 2L, and have measured the total crypt-villus length, finding that the *Tipe0*^{-/-} mice have an increased crypt-villus length (Figure 2O).

We have also assessed activated caspase-3 staining in healthy mice in Figure 2B&C, in addition to the ischemic mice, and see no basal changes between the WT and *Tipe0*^{-/-} mice. This has been noted accordingly in the text (Page 5).

4. Figure 3 and 4. Is difficult to understand the logical behind the experiments presented in both figures. E.g. why the authors analyzed the expression/presence of alkaline phosphatase and keratin 20 in Wt and *Tipe0*^{-/-} organoids? Why the authors used DSS induced colitis as a model in figure 4, given that DSS mainly affects the colon and the experiments in Figure 1 and 2 are in small intestine. Also, a clear connection between the findings reported in figure 3 and 4 and the results reported in figure 1 and 2 is missing. In figure 4, the use of Lgr5 reporter mice instead of Lgr5 antibodies is highly recommended. IF images in figure 4 must be improved. Figure 3A needs to be improved/replaced.

Response: In Figure 3, we characterize the baseline differences in the *Tipe0*^{-/-} intestine that could explain the injury resistance seen. We thought there could be baseline differences given that atypical behavior seen in our enterocyte cultures and proliferation assays, which is why we probed deeper. We have strengthened the explanation of this thought process in the text. Alkaline phosphate and keratin-20 are both markers of differentiated enterocytes, which is part of this analysis. We have edited the text to explain this more clearly (Page 7).

We could not stay in the small bowel because our *Tipe0*^{-/-} mice were resistant to all models of small bowel injury we tested (ischemia and radiation), and thus we could not study post-injury regeneration with this system. From our prior work, we knew that these mice had deficits in recovery from DSS colitis, and so used this system to further probe post-injury regeneration.

In figure 4, Lgr5 antibodies were not used. Rather, this was an RNAScope experiment with a Lgr5 probe, so that we could properly identify the Clu+ cells to measure (as some immune cells also express Clu and these should be excluded); this is per the previously published protocol cited in the manuscript. We have clarified this in the figure legend. Furthermore, we moved the entire RNAScope experiment to Figure E4C&D to space constraints and so we could present larger images in both Figure 4 and Figure 4E.

We have removed Figure 3A, as it was causing confusion. The images for Figure 4 have been improved and are no longer compressed.

5. Figure 5 and 6. The results in figure 1 and 2 are clearly linked to the changes induced in Akt signaling (figures 5 and 6) by the absence of *Tipe0* protein, however the connection established with β -catenin signaling is confusing. TNFAIP8/*Tipe0* absence drives Akt hyperactivation but did not affect Erk 1/2 signaling (Figure 5 and data not shown), the effect of *Tipe0* in Erk must be shown.

Response: Akt activation is known to phosphorylate GSK3 β , which inactivates it and allows for the accumulation of β -catenin. Thus, the changes in β -catenin are a downstream consequence of the changes in Akt activation that we see. We have clarified this point throughout the text (Page 11). We have included the pErk IHC we performed in Figure S1.

Reviewers' Comments:

Reviewer #1:

Remarks to the Author:

The authors revised the manuscript according to the reviewers' comments. However, there still remain significant issues regarding the evaluation of pAktS473 staining, β -catenin staining, and Western blotting of β -catenin.

1. In the revised Figure 5C, the authors claimed that a different Western with more equivalent baselines has been used in the resubmission. It seems that the images for β -catenin and β -actin are replaced using the different sample, however, the images for pAktS473 and Akt remain exactly the same in the revised images as in the previous images (e.g., we can see exactly the same defect in the WT Akt bands in both images). The authors will need to comment on this finding.

Previous Fig 5C

Revised Figure 5C

2. In Fig E5H, β -catenin cannot be detected in the nucleus. Quantification of nuclear β -catenin staining in Fig E5C is not consistent with the staining image in Fig 5H.
3. There still remains issues evaluating the pAkt staining in Fig 5D, because the background staining for *Tipe0^{-/-}* mice are much stronger compared to that of WT mice. The authors claimed that they see this increased staining in the muscularis of *Tipe0^{-/-}* mice by performing IgG control staining. However, this explanation does not justify comparing the

Tipe0^{-/-} and WT mice which have different background staining. The staining strength including the background may also be affected by how the samples are prepared (e.g., how they are fixed, how long they are fixed). We strongly recommend the authors repeat the experiments again to provide more faithful comparison of pAkt staining between Tipe0^{-/-} and WT mice.

Reviewer #2:

Remarks to the Author:

Goldsmith and colleagues have nicely addressed all the comments and concerns raised in the original review. The manuscript is significantly improved, particularly with most of the images that raised a lot of questions now being replaced with higher quality pictures, being removed entirely or modified to better illustrate the main message. This much improved manuscript is now suitable for publication.

Reviewer #3:

Remarks to the Author:

The authors properly addressed the concerns raised in the previous review and significantly improved the manuscript. However, immunofluorescence stainings for E-cadherin must be improved

Point-by-point Responses to the Reviewers

NCOMMS-19-08639C

Thank you very much for giving us this opportunity to respond to the reviewers' comments (shown in blue below). Our responses to the reviewers are delineated below and all changes in the manuscript are marked with red lettering.

Reviewer 1:

In the revised Figure 5C, the authors claimed that a different Western with more equivalent baselines has been used in the resubmission. It seems that the images for β -catenin and β -actin are replaced using the different sample, however, the images for pAktS473 and Akt remain exactly the same in the revised images as in the previous images (e.g., we can see exactly the same defect in the WT Akt bands in both images). The authors will need to comment on this finding.

Response: As we wrote in our last rebuttal, “a *different Western with more equivalent baselines has been used in the resubmission.*” We never meant to imply entirely new data was used for the entire figure, but rather that we used a different Western image for the β -catenin portion of the experiment, which was the portion of the data in question. We have run the experiment multiple times and with multiple exposures, and so chose a different β -catenin gel image to use. In this process, we realized that the β -actin image had been flipped inadvertently in the initial submission, and so we flipped it back, as the editors noted and has been addressed in the document entitled “Response to the Editors.”

In Fig E5H, β -catenin cannot be detected in the nucleus. Quantification of nuclear β -catenin staining in Fig E5C is not consistent with the staining image in Fig 5H.

Response: We attempted to quantify nuclear staining as best we could, at the reviewer's previous request. The overlap of the Hematoxylin with the DAB chromagen makes this difficult, especially given the strong cytoplasmic staining that draws the eye, but we feel we can appreciate nuclear staining, especially on the magnified images. However, the difficulty in appreciating nuclear staining is why we used RNASeq to demonstrate that β -catenin signaling was induced and thus active/nuclear. Personally, we are not attached to the manual quantification of the IHC nuclear β -catenin staining, as we think the RNASeq is sufficient, and would be comfortable including or removing it as the editor sees fit. As suggested by the editor, we have moved the IHC β -catenin nuclear staining data to Figure S1f and adjusted the figure legends and text accordingly. The body of the text already focused on the scRNASeq data to make the associated point, so the main body of the manuscript has only minor edits.

There still remains issues evaluating the pAkt staining in Fig 5D, because the background staining for Tpe0^{-/-} mice are much stronger compared to that of WT mice. The authors claimed that they see this increased staining in the muscularis of Tpe0^{-/-} mice by performing IgG control staining. However, this explanation does not justify comparing the

Tipe0^{-/-} and WT mice which have different background staining. The staining strength including the background may also be affected by how the samples are prepared (e.g., how they are fixed, how long they are fixed). We strongly recommend the authors repeat the experiments again to provide more faithful comparison of pAkt staining between Tipe0^{-/-} and WT mice.

Response: The reviewer suggests redoing the staining in its entirety. This was already done and presented in Figure 5E, with the control column of Figure 5E matching the healthy column of Figure 5D. All samples were fixed for identical amounts of time, and we have repeated the experiment several times, often as controls for other experiments (specifically the radiation and 60' ischemia with no reperfusion studies). In some cases, we have repeated the staining for each experiment multiple times as well. We also see the same staining pattern after radiation (Figure E5D). We do not think further repeats are warranted, given the number of times it has already been repeated. As a result, per the editor's suggestion, in Figure 5d, we have selected new representative images for the WT and *Tipe0^{-/-}* healthy mouse pAKT staining to better match what is seen in Figure 5e and to ensure consistency in both panels; we also adjusted the white balance of the ischemic images so they would be consistent with the new pair of healthy controls. As a result of these changes, we have removed the IgG data from Figure S1f, as this is now unnecessary.

Reviewer #3:

The authors properly addressed the concerns raised in the previous review and significantly improved the manuscript. However, immunofluorescence stainings for E-cadherin must be improved.

Response: We were able to adjust the coloring of the merged image for Figure 4c to increase the visibility of both channels, but not for Figure 4e. The images we have available for Figure 4e are the best we have access to for the Sca-1/E-cadherin staining contrast given the current situation with the coronavirus pandemic; we cannot take new images at this time. Thus, at the editor's suggestion, we have opted to make it clearer both in the figure legend and the text that the E-cadherin is a counterstain to identify the enterocyte-specific Sca-1 signal.